# Susceptibility profile of *Anopheles* and target site resistance mechanism against organophosphates in Cameroon.

Judith Dandi-Labou[1]*, Jonas A. Kengne-Ouafo[1], Leon Mugenzi[1], Magellan Tchouakui[1], Murielle Wondji[1], Charles S. Wondji[1,2,3]*

**1** Centre for Research in Infectious Diseases, Yaoundé, Cameroon, **2** Vector Biology Department, Liverpool School of Tropical Medicine, Pembroke Place, Liverpool, United Kingdom, **3** International Institute of Tropical Agriculture, Yaoundé, Cameroon

* judith.labou@cridcam.net (JD-L); charles.wondji@lstmed.ac.uk (CSW)

## Abstract

Escalating pyrethroid resistance in malaria vectors jeopardizes vector control, necessitating the use of alternative insecticides such as pirimiphos-methyl (organophosphate) for indoor residual spraying (IRS). Tracking the spread of resistance and elucidating its molecular basis are essential for effective resistance management against these insecticides. This study monitored resistance to two organophosphates, malathion (MA) and pirimiphos methyl (PM), in three malaria vectors (*Anopheles gambiae* s.s., *An. coluzzii*, and *An. funestus* s.s.) across Cameroon and explored related resistance mechanisms. WHO tube assays revealed that *An. funestus* s.s. populations were fully susceptible to both organophosphates; *An. coluzzii* populations were either fully susceptible (North) or potentially resistant (South; 97% mortality). In contrast, the two *An. gambiae s.s.* populations in this study were resistant: in the rural agricultural hotspots of Mangoum (94% mortality to PM; 50% to MA) and in peri-urban cultivated location of Nkolondom, which exhibited the highest resistance to both PM (80% mortality) and MA (46% mortality), associated with recorded use of organophosphates by farmers. Genotyping the Ace-1 markers revealed a close association with susceptibility profile, as no resistance allele was observed in *An. funestus* s.s. and in the northern population of *An. coluzzii* and a very low frequency in Njombe (3%). In contrast, a higher frequency of Ace-1$^R$ was observed in *An. gambiae* s.s. with a significant association observed with resistance (PM: OR = 20.33, P = 0.04; MA: OR = 98.33, P = 0.0019). Furthermore, analysis of 100 Ace-1 clones showed copy number variation was linked to resistance, as resistant mosquitoes displayed higher copy numbers compared to susceptible individuals. These findings suggest that malaria control with organophosphate-based IRS is a viable alternative in Cameroon; however, it will be necessary to consider the distribution of species and the development of resistance.

**Data availability statement:** All the direct sequences files are available from the GenBank database (accession number(s) PQ602501, PQ602517.)

**Funding:** This research was funded by a grant from the Bill & Melinda Gates Foundation (Grant INV-006003) awarded to CSW. The views presented in the manuscript are those of the authors and do not necessarily reflect the views of the BMGF.

**Competing interests:** The authors have declared that no competing interests exist.

## Introduction

Africa bears a disproportionately high share of the global malaria burden, with 94% of the 263 million cases estimated worldwide in 2023 occurring on the continent [1]. Cameroon, classified as one of the 11 most affected countries, recorded an estimated 7.3 million cases that same year [1]. Malaria control efforts in Africa heavily rely on insecticide-based interventions, primarily long-lasting insecticidal nets (LLINs) and indoor residual spraying (IRS). These intervention strategies have contributed to > 70% of the decline in malaria burden [2]. During the past decades, the World Health Organization (WHO) recommended four main chemical classes of insecticides for vector control: organochlorines, pyrethroids, organophosphates, and carbamates [3]. While pyrethroids currently dominate in LLINs due to their high mosquito killing efficacy and low mammalian toxicity, the widespread resistance of mosquito vectors to this class of insecticide poses a growing challenge to malaria control.

To counter this challenge, recommendations include rotating or combining insecticides with different modes of action, exploring new classes, and implementing diverse control interventions [4]. Pirimiphos-methyl, a broad-spectrum organophosphate insecticide with contact and fumigant action, offers such alternative, having found success against pyrethroid-resistant mosquitoes [5–8]. This widely used insecticide in agriculture, has been recommended by the WHO for IRS, either in rotation or in combination with other methods [9]. Although there are consistent reports of greater susceptibility of mosquitoes to pirimiphos-methyl across the continent compared to other insecticides within these classes [10], unfortunately, concerns have been raised regarding emerging resistance to this insecticide as well as to other organophosphates (OPs), including fenitrothion, malathion, and to carbamates (CXs) in West and East African populations [10–14].

Both CXs and OPs work by blocking the action of the acetylcholinesterase enzyme (AChE), encoded by the Ace-1 gene in mosquitoes by competitively binding to the AChE's active site. This competitive binding prevents AChE from breaking down acetylcholine (Ach), a neurotransmitter crucial for normal nerve function [15]. As a result, Ach accumulates, leading to uncontrolled muscle stimulation due to continuous nerve impulses, conducting to the death of the insects [15].To survive exposure to these insecticides, mosquitoes have developed mutations within the Ace-1 gene. These mutations alter the structure of the AChE enzyme, making it less susceptible to binding by CXs and OPs. The single amino acid change from glycine to serine mutation in codon 280 (Ace-1 G280S, previously known as Ace-1 G119S) and copy number variants (CNVs) increasing Ace-1 gene copies in the genome of *An. gambiae* s.l. are known to be linked to resistance against these insecticide classes [11,14,16]. Similarly in *An. funestus* s.s., the mutation Ace-1 N485I have been described to be involved in CXs resistance [17].

Resistance to pirimiphos-methyl has been reported in areas where IRS is recommended for vector control [13,14,18]. However, the use of organophosphates for IRS so far has not yet been implemented in some countries such as Cameroon where pyrethroid-based interventions are predominant [19,20]. Due to the increased levels of pyrethroid resistance reported in such countries, IRS could

be recommended as an alternative besides next-generation control tools to tackle pyrethroid-resistant mosquito populations. However, with the increasing use of OPs in agricultural settings, it is imperative to continuously monitor the presence and the extent of mosquitoes' organophosphates resistance in the field, especially insecticides recommended by WHO for IRS.

This study aimed to assess the resistance levels of three major malaria vectors *An. gambiae* s.s., *An. coluzzii*, and *An. funestus* s.s. against two organophosphates: pirimiphos-methyl (PM) and malathion (MA) across Cameroon to advise on the alternative in using PM for IRS campaign by NMCP and its partners.

## Materials and methods

### Mosquito sampling

Six localities across five geographic regions in Cameroon were chosen based on the presence of different *Anopheles* species (Fig 1A). The study was conducted from April 2021 to January 2022 across these six sampling sites (Fig 1B). Mosquitoes at the larval and pupal stages were collected using "dipping" method [21] in Nkolondom, where *An. gambiae* s.s. immature stages were abundant. Indoor aspiration method on adult mosquitoes was used in Elende, Mibellon, Njombe and Gounougou where *An. funestus* s.l. and *An. gambiae* s.l. breeding sites had low densities of immature stages. Both techniques were carried out in Mangoum.

Collected immature stages were reared to adulthood at the Center for Research in Infectious Diseases (CRID) insectary in Yaoundé. Adult mosquitoes collected by indoor aspiration, once fully gravid, were individually placed in 1.5 ml microcentrifuge tubes for egg-laying and larvae reared to F1 adults. All mosquitoes were morphologically identified using standard keys of Gillies and De Meillon [22] and Gillies and Coetzee [23].

QGIS version 3.28.3 was used to generate the map using open access sharefiles (https://gadm.org/)"

This study is a part of a project that received ethical clearance from the National Research Ethics Committee for Human Health, registered under number 2021/07/1372/CE/CNERSH/SP to ensure the safety of study participants during the collection of adult mosquitoes. Furthermore, permissions were obtained from village chiefs and homeowners at the locations where mosquitoes were collected.

### Insecticide resistance profile

Insecticide susceptibility bioassays were performed following WHO protocols [24]. Two- to five-day-old F0 or F1 adult female mosquitoes were exposed to impregnated papers at discriminating concentrations (0.25% PM, 5% MA), and untreated (control) for 1 hour. All the tests were performed in insectary conditions of 25±1°C temperature and 70–80% relative humidity. Mortality rates were recorded after 24 hours, and resistance status was evaluated according to WHO criteria [24]. Alive and dead (mosquitoes that survived/died after test exposure) mosquitoes were stored at -20°C for further molecular analysis.

### DNA extraction and species identification

Genomic DNA were extracted from the whole mosquitoes following the Livak protocol [25]. Cocktail PCR method was used to distinguish members of the *An. funestus* group as previously described [26] whereas Short-Interspersed Elements (SINE) PCR assay [27] was used to identify members of the *An. gambiae* complex. Our approach to species identification varied based on the expected composition of local vector populations. In Njombe with known coexisting vector species (sympatric populations) within the same complex [28], we performed species identification on all individual mosquitoes that underwent WHO susceptibility tests. However, in Mangoum, Nkolondom, Elende, Mibellon, and Gounougou, previously known to have a single dominant vector species or morphologically identifiable populations [29–33], we tested only a subset of mosquitoes to confirm the species composition.

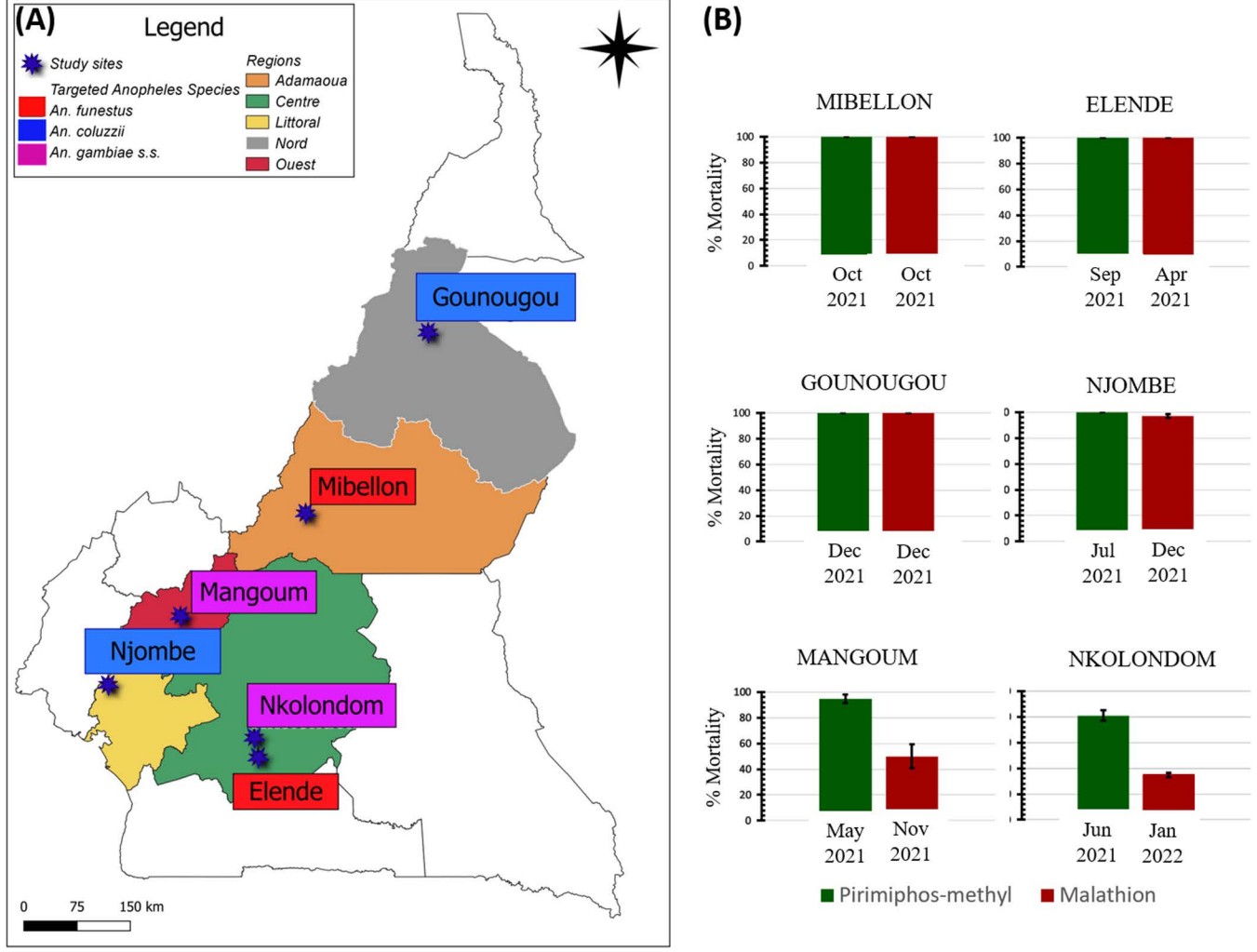

**Fig 1. Study sites.** (A) Geographical representation of collection sites in Cameroon. The study sites are represented by stars. The color represents the targeted species. (B) Susceptibility profile using WHO bioassay tube test according to the study sites. Mortality rates were recorded 24 h post-exposure to the discriminating concentration of insecticides. Data are shown as mean±standard error. Mosquitoes were sampled both pirimiphos-methyl and malathion tests on the same day at Gounougou (December 2021) and Mibellon (October 2021). For all other mosquito populations, the sampling schedule differed.

## TaqMan genotyping of Ace-1 G280S and N485I mutations

The presence of G280S mutation in *An. gambiae* s.l. and N485I mutation in *An. funestus* s.l. was investigated using TaqMan real-time PCR assays on an Agilent Mx3005 qRT-PCR thermocycler following the protocols described by Bass and colleagues [34] and Ibrahim et colleagues [17] respectively with minor modifications. Each reaction was conducted in 10 µl total volume containing 5 µl SensiMix (Bioline, London, UK), 0.25 µl of 40x Probe Mix coupled to allelic-specific primers for *An. gambiae* s.l. [Ace-1 forward (5′-GGC CGT CAT GCT GTG GAT-3′); Ace-1 reverse (5′-GCG GTG CCG GAG TAG A-3′); ACE1-HEX (5′-TTC GGC GGC GGC T-3′); ACE1–6-FAM (5′-TTC GGC GGC AGCT-3′)], or for *An. funestus* s.l [Ace-1 forward (5′- CAT GCG ATA CTG GTC AAA CTT TGC -3′); Ace-1 reverse (5′- GCC ATT CGG GAA ATT CGC TAC TA-3′); ACE1-HEX (5′- CAA ACC CCA ACA CGG C -3′); ACE1–6-FAM (5′- CAA ACC CCA TCA CGG C -3′)], 4.25 µl

of dH$_2$0, and 1 µl of genomic DNA. Thermal cycling conditions were the following: an initial denaturation 10 min at 95°C, followed by 40 cycles each of 92°C for 15 s and 60°C for 1 min. Two probes labeled with fluorochromes FAM™ and HEX™ were used to detect respectively the resistant mutant Ace-1$^R$ and the wild-type susceptible alleles Ace-1$^S$. At the end of amplification, genotypes were scored from bi-directional scatter plots of results produced by the Mx3005 v4.10 software. The association between G280S genotypes and resistance phenotypes was assessed using odds ratios (OR) calculated in Medcalc software https://www.medcalc.org/calc/odds_ratio.php.

### Ace-1 amplification and sequencing

To confirm TaqMan genotypes and investigate polymorphisms in the Ace-1 gene, a 924 bp region encompassing exons 4–6 (VectorBase AgamP4 annotation, AGAP001356; G280S position in exon 5) was amplified from 20 mosquitoes exposed to pirimiphos-methyl at Nkolondom. TaqMan-genotyped alive and dead mosquitoes were selected for PCR amplification.

PCR amplification was carried out following the protocol previously described by Essandoh and collaborators [11]. Briefly, a 50 µl reaction mix containing 10 pml each of primers Ex2Agdir1 (5'-AGG TCA CGGTGA GTC CGTACG A-3') and Ex4Agrev2 (5'- AGG GCG GAC AGC AGA TGC AGC GA -3'), 10 mM dNTPs, ddH2O, 5X HF Phusion buffer, and 1 U Phusion Taq polymerase was set up with the following cycling conditions: initial denaturation at 98°C for 3 min, followed by 35 cycles of 98°C for 30 sec, 64°C for 30 sec, and 72°C for 30 sec, with a final extension at 72°C for 10 min. Three (03) µl of each amplicon were visualized on 1.5% agarose gel stained with MIDORI Green. The rest of the PCR products were purified using the ExoSAP-IT™ PCR cleanup reagent (ThermoFisher Scientific, Waltham, MA, USA) and then directly sequenced commercially using the same primers Ex2Agdir1 and Ex4Agrev2 to confirm the presence of the G280S mutation and assess selection signatures. DNA sequences were aligned with a susceptible *An. gambiae* s.s. reference from VectorBase (gene ID: AGAP001356) using ClustalW [35] implemented in BioEdit [36]. Polymorphism analysis was performed with DnaSP v5.10 [37], and a maximum-likelihood tree was constructed from aligned sequences in MEGA 10.1.0 [38] using default parameters of Tamura 3 model. Additionally, a haplotype network was built using NETWORK software (version 10.1.0) from Fluxus Engineering (http://www.fluxus-engineering.com).

### Cloning to assess Ace-1 duplication

Fifteen mosquitoes exposed to pirimiphos-methyl (5 alive, 5 dead heterozygotes, 5 dead susceptible) were chosen for investigating Ace-1 duplication. Additionally, 5 unexposed Kisumu (susceptible lab strain) mosquitoes were included. DNA from each sample was amplified and purified using a Qiaquick purification kit (Qiagen, Hilden, Germany). The amplified fragments were cloned using a CloneJET™ PCR Cloning Kit (Thermo Scientific, Waltham, MA, USA). Colonies were screened for inserted amplicons using pJET1.2 primers (manufacturer's instructions). Bands around 1650 bp (expected size for Ace-1 in pJET vector) were considered as potential clones. For each individual, ~5 clones were amplified, purified using a MiniPrep kit (Qiagen), and Sanger sequenced. All successfully sequenced samples were aligned, and polymorphism analysis was performed as described above.

### qPCR to detect copy number variation (CNV)

The concentrations and purity of extracted DNA of the 5 alive, 5 dead heterozygotes, 5 dead susceptible and 5 Kisumu were measured using a NanoDrop spectrophotometer. Dilutions were performed to standardize all samples to 5 µg/µl before amplification by qPCR.

Quantitative polymerase chain reaction (qPCR) was performed to investigate the presence of copy number variations in the Ace-1 gene. Custom primers (Ace1duplF1: 5'-TTACTCTCAGCAAGGACGCA-3'; Ace1duplR1: 5'- TGGAGTCAGACGATGGAACC-3') were designed to quantify the relative number of Ace-1 copies in each DNA sample. All reactions were run alongside the housekeeping gene ribosomal protein S7 (RSP7) for normalization purposes.

Each reaction mixture contained 1 µl of genomic DNA, 0.6µM of specific primers (Ace-1dupl or RSP7) and 10µl of 2X Brilliant II SYBR® Green master mix (Agilent). The reactions were run on an MX3005 real-time PCR system (Agilent) following a dissociation curve made by 10 min for 95°C, 40 cycles of 10 s at 95°C and 60°C respectively, with melting curves run after each end point amplification at 1 min for 95°C, followed by 30 s increments of 1°C from 55°C to 95°C. The $2^{-\Delta\Delta CT}$ method was then used to estimate the number of Ace-1 copies in each group relative to the Kisumu population (chosen as the reference susceptible group). Finally, a Kruskal-Wallis non-parametric test was used to assess statistically significant differences in copy number between groups.

## Results

### Species identification

The species identification confirmed that *An. funestus* s.s. constituted the main vector species in Elende and Mibellon, while *An gambiae* s.s constituted the main species in Mangoum and Nkolondom. In Njombe, around 10% of mosquitoes tested belonged to *An. gambiae* s.s., with the remaining 90% being *An. coluzzii*. Gounougou had only *An. coluzzii* (S1 Fig).

### Insecticide resistance profile

A total of 1200 adult female mosquitoes (around 100 per location per insecticide) belonging to *An. gambiae* s.s. (33.33%), *An. coluzzii* (33.33%) and *An. funestus* s.s. (33.33%) from the six study sites were exposed to both pirimiphos-methyl and malathion. Both populations of *An. funestus* s.s. (Mibellon and Elende) were susceptible to both pirimiphos-methyl and malathion insecticides (Fig 1B). *An. coluzzii* mosquitoes from Gounougou were also susceptible to both insecticides. In Njombe, *An. coluzzii* displayed potential resistance to malathion (97% mortality) but remained susceptible to pirimiphos-methyl. *An. gambiae* s.s. population from Mangoum showed possible resistance to pirimiphos-methyl (94% mortality) and was resistant to malathion (50% mortality). Alarmingly, mosquitos populations from Nkolondom exhibited greater resistance level to PM (80% mortality) and MA (46% mortality).

### TaqMan genotyping of Ace-1 G280S and N485I mutation

In *An. funestus* s.s. from Elende, Mibellon, and Gounougou, 30 mosquitoes exposed to each insecticide were genotyped. Similarly, 58 and 71 mosquitoes were genotyped in Nkolondom after exposure to pirimiphos-methyl and malathion, respectively. Additionally, 48 mosquitoes exposed to malathion were genotyped in Njombe (among which two *An. gambiae* s.s), while 36 unexposed females were genotyped in Mangoum.

Genotyping of G280S and N485I mutations revealed the absence of resistant alleles for the N485I Ace-1 mutation in all *An. funestus* s.s. populations, supporting their full susceptibility to CXs and OPs. The G280S resistant allele was detected in *An. gambiae* s.l. mosquitoes from Nkolondom (86%), Mangoum (53%), and Njombe (3%) (Fig 2). The higher frequency of Ace-1R in Nkolondom correlates with the greater resistance to PM observed in this location.

### Correlation between genotypes and phenotypes

From the 71 individuals exposed to PM in Nkolondom, 61 (85,9%) exhibited the homozygous resistant allele (RR), 10 (14,1%) exhibited the homozygous susceptible allele (SS), and no heterozygous genotype (RS) was found in this population (Fig 3). Thirty of the 61 RR mosquitoes (49.2%) were alive, whereas all 10 SS mosquitoes were dead. This translates to a 100% frequency of the Ace-1R allele among surviving individuals. Mosquitoes with Ace-1R allele displayed a significantly higher ability to survive compared to those with the Ace-1S allele (OR = 64.37, 95% CI = 3.85–1075.38; P < 0.003). This correlation was further confirmed when comparing the only two genotypes obtained, RR and SS (OR = 20.33, 95% CI = 1.14–362.38; P < 0.04).

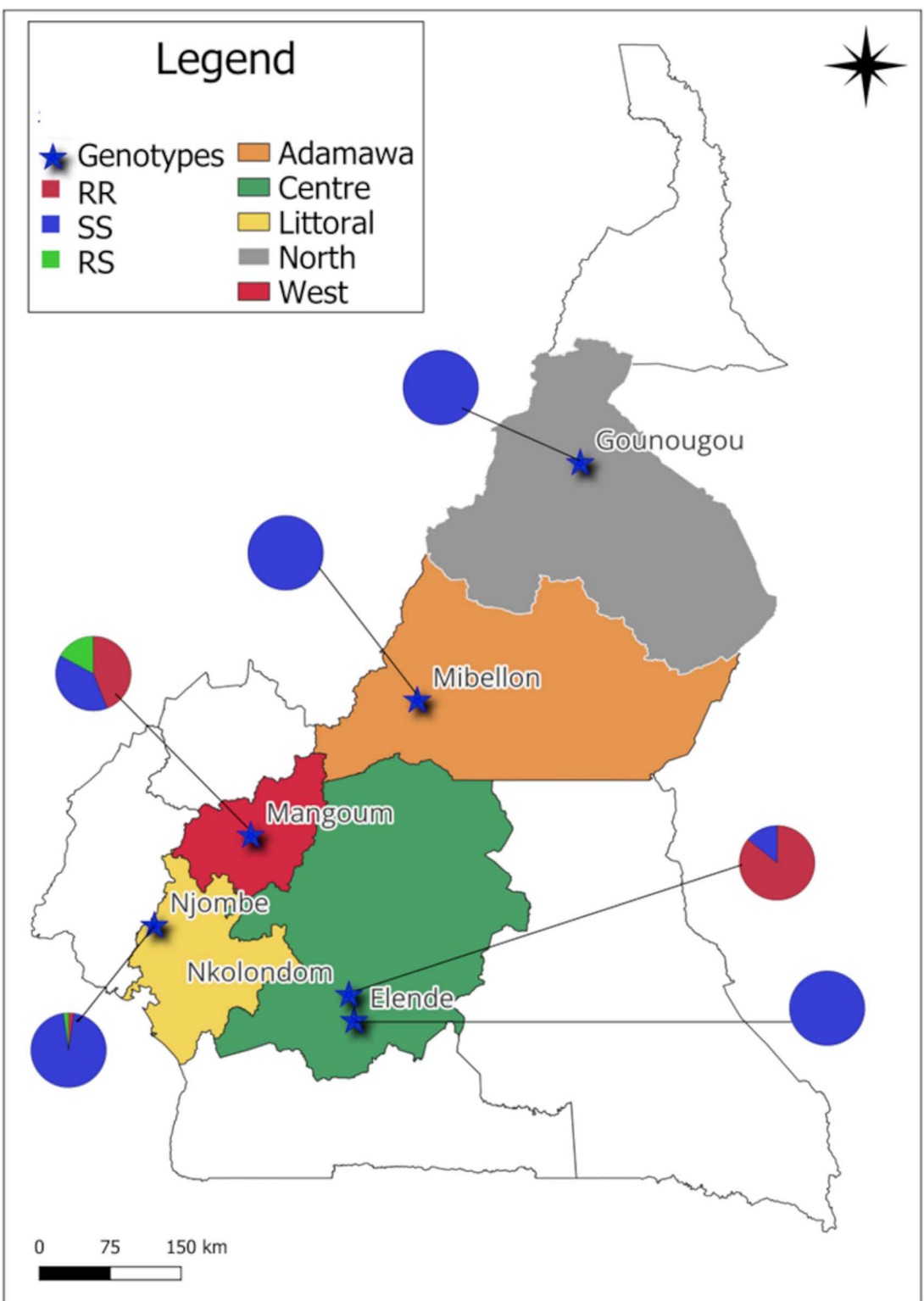

**Fig 2. Distribution of Ace-1 genotypes in major *Anopheles* species in Cameroon.** Mosquitoes are the batch exposed to PM at Nkolondom, Elende and Gounougou, exposed to MA at Njombe, and non-exposed at Mangoum. The alleles frequencies are given in %.

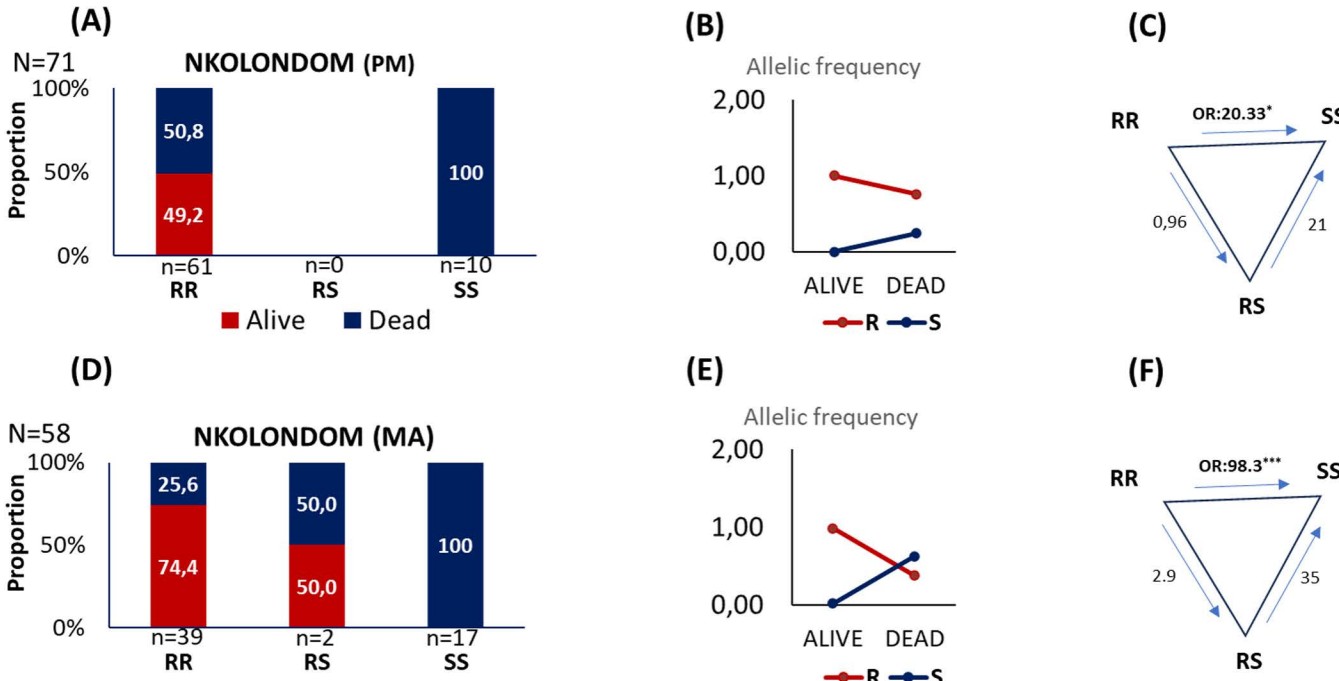

**Fig 3. Proportion of Ace-1 genotypes, allelic frequencies and Odds Ratio.** The results are those of mosquitoes from Nkolondom exposed to PM (A, B and C) and exposed to MA (D, E and F).

A similar trend was observed in Nkolondom mosquitoes exposed to malathion (MA). Thirty out of the 39 RR mosquitoes (74.4%) were alive, while only one of the two heterozygous (RS) individuals (50%) died. All 17 SS mosquitoes (100%) were dead. This resulted in a 98% frequency of the Ace-1$^R$ allele in the surviving mosquito population. Mosquitoes with the Ace-1$^R$ allele again showed significantly higher survival rates compared to those with the Ace-1$^S$ allele (OR = 79.94, 95% CI = 18.62–343.24; P < 0.0001). The correlation between survival and genotype was also significant when comparing RR vs SS (OR = 98.33, 95% CI = 5.42–1783.68, P < 0.0019), but not when comparing RR vs RS (OR = 2.9, 95% CI = 0.16–50.81, P < 0.46) or RS vs SS (OR = 35, 95% CI = 0.94–1292.51, P < 0.053).

In the Njombe population exposed to malathion, the lone survivor (2.2%) displayed the RR genotype. One dead mosquito (2.2%) was heterozygous (RS), and the remaining 44 (95.6%) were homozygous susceptible (SS).

### Ace-1 gene polymorphism analysis

**Confirmation of genotyping result by direct sequencing.** TaqMan genotyping of mosquitoes from Nkolondom exposed to pirimiphos-methyl identified only two genotypes for the Ace-1 gene: RR and SS. To confirm these genotypes, 20 mosquitoes were selected for direct sequencing: 10 alive TaqMan-RR, 5 dead TaqMan-RR, and 5 dead TaqMan-SS. Sequencing of the alive TaqMan-RR individuals confirmed the presence of the homozygous RR-Ace-1 genotype in nine mosquitoes, while one individual was identified as heterozygous (Ace-1$^{RS}$) with the presence of two peaks corresponding to A and G nucleotides in the chromatogram (S2 Fig). Among the dead TaqMan-RR mosquitoes, three mosquitoes were heterozygous (RS) while two sequences were unusable due to overlapping peaks (Figure **a** in S3 Fig). Sequencing of the dead TaqMan SS individuals confirmed this genotype in four mosquitoes, with one sequence being unusable due to overlapping peaks (Figure **b** in S3 Fig).

S4 Fig shows the scatter plot from the TaqMan PCR analysis, highlighting the positions of individual mosquitoes chosen for sequencing within the RR and SS clusters. This analysis shows that, despite the overall positive correlation (80%)

between the two methods $R^2 = 0.8086$ $P = 0.000086$ (Spearman correlation test), TaqMan genotyping of Ace-1 is less sensitive to differentiate heterozygous, probably due to the known duplication events within this gene.

Furthermore, to investigate Ace-1 mutations in *Anopheles* individuals from Njombe exposed to malathion, six mosquitoes were chosen for sequencing: 3 alive TaqMan RR mosquitoes (2 *An. gambiae* s.s., 1 *An. coluzzii*), 1 dead RS (*An. coluzzii*), and 2 dead SS (01 *An. gambiae* s.s., 01 *An. coluzzii*). Two of these sequences were unusable due to overlapping peaks (S5 Fig). The two *An. gambiae* s.s. alive TaqMan RR, and the *An. coluzzii* dead TaqMan RS harbored the heterogenous peak A/G, confirming the presence of Ace-1$^R$ allele circulating in each of the two sympatric species. The dead *An. coluzzii* TaqMan SS was confirmed to be homozygous susceptible.

**Genetic diversity analysis.** Analysis of the polymorphism patterns of the Ace-1 portion in *An. gambiae* s.s. from Nkolondom resulted in the alignment of a common 926 bp, leading to the detection of 32 polymorphic sites in mosquitoes (Table 1). A consistent reduced diversity was observed in resistant mosquitoes than in susceptible ones with lower number of substitution sites (n = 6 vs 32), lower number of haplotypes (**h = 2 vs 12**), lower haplotype diversity (**Hd = 0.1 vs 0.978**), and the nucleotide diversity (**π = 0.0006 vs 0.0104**). The single non-synonymous substitution detected was the G280S mutation. However, in dead mosquitoes 18 synonymous mutation was observed, with an important rate coming from dead SS. Tajima's D and the Fu and Li index (**D***) were negative and statistically significant in alive phenotype only further supporting the signature of selection.

Concerning haplotypes, twelve out of 13 were recorded in dead mosquitoes, shared between dead RS and dead SS with 5 and 7 haplotypes respectively. Among the alive mosquitoes two haplotypes were recorded, one from Ace-1$^R$ allele and the second from the unique Ace-1$^S$ allele, provided by the heterozygote alive individual. The most dominant haplotype containing-Ace-1$^R$ allele was H1 (Fig 4A) found in alive mosquitoes as well as some dead heterozygous. The haplotype network (Fig 4B) showed a trend of clustering according to genotype, with all Ace-1$^{SS}$ haplotypes (formed by dead SS mosquitoes) grouped in one cluster and the Ace-1$^{RR}$ and Ace-1$^{RS}$ in another cluster. Furthermore, the phylogenetic tree (Fig 4C) emphasized the previous observation by clearly showing the two main clusters.

**Cloning to assess for duplication.** To further elucidate the interaction between Ace-1$^R$ and resistance phenotypes, we assessed the presence of Ace-1 duplication and the copy number variation. The presence of more than two haplotypes within clones from an individual indicated duplication, as a diploid genome without duplication would have

**Table 1. Statistics for polymorphism in Ace-1 gene including the G280S mutation in dead and alive *An. gambiae* s.s. mosquito populations from Nkolondom after exposure to pirimiphos-methyl.**

| | 2N | S | h | Hd | π | Syn | NSyn | D | D* |
|---|---|---|---|---|---|---|---|---|---|
| **ALIVE** | 20 | 6 | 2 | 0.1 | 0.0006 | 1 | 1 | -2.05624 * | -2.95983** |
| **DEAD** | 14 | 32 | 12 | 0.978 | 0.0104 | 18 | 1 | -0.40350 ns | -0.35016 ns |
| **ALIVE RS** | 2 | 6 | 2 | 1 | 0.0064 | 1 | 1 | n.aª | n.aª |
| **ALIVE RR** | 18 | 0 | 1 | 0.0 | 0.0 | 0 | 0 | n.a$^b$ | n.a$^b$ |
| **DEAD RS** | 6 | 7 | 5 | 0.933 | 0.0036 | 1 | 1 | 0.50809 ns | 0.29105 ns |
| **DEAD SS** | 8 | 22 | 7 | 0.964 | 0.0074 | 15 | 0 | -1.14773 ns | -0.92481 ns |
| **ALIVE & DEAD RS** | 26 | 7 | 6 | 0.354 | 0.0015 | 1 | 1 | -0.73001 ns | 0.61011 ns |
| **ALIVE & DEAD SS** | 28 | 31 | 9 | 0.545 | 0.0076 | 18 | 1 | -0.61083 ns | -0.83044 ns |
| **OVERALL** | 34 | 32 | 13 | 0.6239 | 0.0070 | 18 | 1 | -0.77480 ns | -1.1318 ns |

*2N* number of sequences, *S* number of polymorphic sites, *h* number of haplotypes, *Hd* haplotype diversity, π nucleotide diversity *Syn* synonymous substitution, *NSyn* non-synonymous substitution, *D* Tajima's statistics, *D*

*Fu and Li's statistics (the asterisk indicates without an outgroup), *ns* not significant,

*:P<0.05,

**:P<0.02, *n.aª* not applicable because need at least 4 samples to compute. *n.a$^b$* not applicable because need polymorphism to compute.

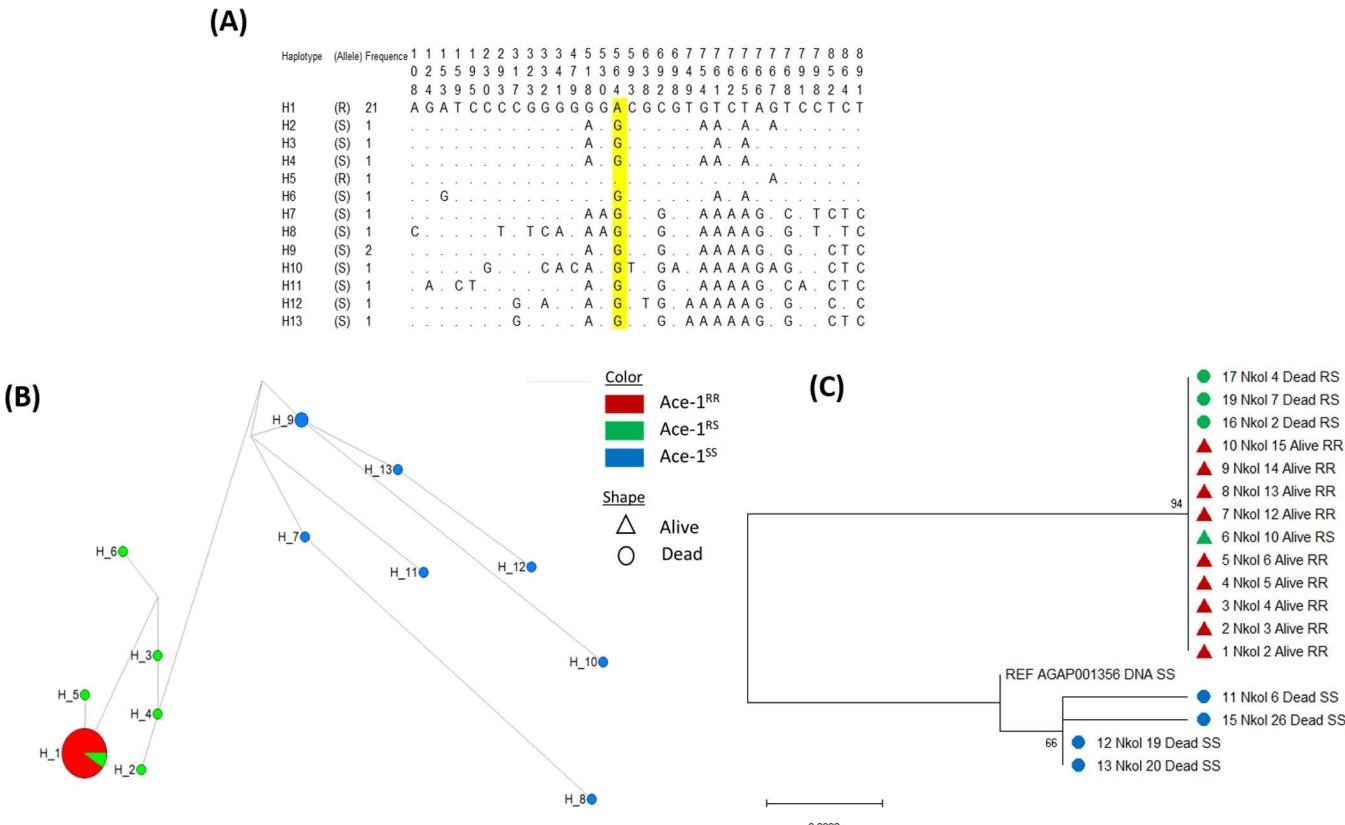

**Fig 4. Polymorphism patterns of Ace-1 gene from direct sequencing.** (A) Polymorphic sites and haplotypes detected. Haplotypes are labelled with S (susceptible) or R (resistant) and the frequency is provided. (B) Haplotype network shows the links among resistant (Red), heterogenous (green) and susceptible haplotypes (blue). (C) Maximum-likelihood phylogenetic tree of Ace-1 gene supporting the clustering of haplotypes according to the mosquito genotypes.

a maximum of two haplotypes. Table 2 presents the total number of clone sequences analyzed per individual in each phenotype group. Out of the five clones from the alive RS individual (individual 6Nkol on Table 2), 4 had Ace-1$^R$ allele and one Ace-1$^S$ allele. Interestingly, while expecting two haplotypes, they presented four different haplotypes. All alive RR individuals gave clones with Ace-1$^R$ allele, and they showed a maximum of two haplotypes. Among clones from dead RS individuals, we found the two different Ace-1 alleles (Ace-1$^R$; Ace-1$^S$), and more than two haplotypes in some individuals, demonstrating the presence of duplication among dead RS mosquitoes. Moreover, this cloning helps to confirm the heterogenous status of the 2 previously sequences that were discarded from direct sequencing analysis (18Nkol and 20 Nkol), showing that all the five dead TaqMan RR individual were heterozygous for the duplication. Dead SS and Kisumu clones harbored only the susceptible allele. Overall, most of dead mosquitoes, as well as Kisumu had more than two haplotypes. As for direct sequencing on the dead and alive TaqMan individuals, the phylogenetic tree made with clones (Fig 5) showed two main clusters made of Ace-1$^{RR}$ and Ace-1$^{RS}$ on one hand and Ace-1$^{SS}$ on the other hand. Two clones of dead RS individuals (16 Nkol RS clone 4 and 17 Nkol RS clone 3) clustered more with the ACE-1$^{SS}$ cluster, as well as 3 clones of the alive RS individual.

**qPCR to detect Copy Number Variation (CNV).** To confirm the presence of duplications in our study population, qPCR was carried out to detect CNV. Alive mosquitoes had a significantly higher number of Ace-1 gene copies (2.4±0.5) compared to dead SS (0.8±0.4) mosquitoes and the susceptible reference group Kisumu (1±0.2) (Fig 6A). No significant

**Table 2. Total number of clone sequences analyzed per individual in each phenotype group. The number of different Ace-1 allele found and the number of haplotypes per individual is provided.**

| Phenotype | Individuals | Genotype (Sanger) | Number of Clone | Alleles found among clones | Number of haplotype |
|---|---|---|---|---|---|
| Alive | 6 Nkol | RS | 5 Clones | 4 ACE-1$^R$; 1 ACE-1$^S$ | 4 |
| | 7 Nkol | RR | 5 Clones | 5 ACE-1$^R$ | 1 |
| | 8 Nkol | RR | 5 Clones | 5 ACE-1$^R$ | 1 |
| | 9 Nkol | RR | 5 Clones | 5 ACE-1$^R$ | 2 |
| | 10 Nkol | RR | 5 Clones | 5 ACE-1$^R$ | 1 |
| Dead | 16 Nkol | RS | 5 Clones | 4 ACE-1$^R$; 1 ACE-1$^S$ | 2 |
| | 17 Nkol | RS | 5 Clones | 5 ACE-1$^R$ | 2 |
| | 18 Nkol | RS | 5 Clones | 2 ACE-1$^R$; 3 ACE-1$^S$ | 4 |
| | 19 Nkol | RS | 5 Clones | 5 ACE-1$^R$ | 3 |
| | 20 Nkol | RS | 4 Clones | 3 ACE-1$^R$;1 ACE-1$^S$ | 3 |
| | 11 Nkol | SS | 5 Clones | 5 ACE-1$^S$ | 2 |
| | 12 Nkol | SS | 5 Clones | 5 ACE-1$^S$ | 3 |
| | 13 Nkol | SS | 5 Clones | 5 ACE-1$^S$ | 5 |
| | 14 Nkol | SS | 5 Clones | 5 ACE-1$^S$ | 3 |
| | 15 Nkol | SS | 5 Clones | 5 ACE-1$^S$ | 3 |
| Susceptible Lab strain | 1 KIS | SS | 4 Clones | 4 ACE-1$^S$ | 1 |
| | 2 KIS | SS | 5 Clones | 5 ACE-1$^S$ | 4 |
| | 3 KIS | SS | 4 Clones | 4 ACE-1$^S$ | 2 |
| | 4 KIS | SS | 6 Clones | 6 ACE-1$^S$ | 3 |
| | 5 KIS | SS | 5 Clones | 5 ACE-1$^S$ | 5 |

difference in copy number was observed between alive and dead RS mosquitoes or between dead SS mosquitoes and the Kisumu. In addition, Spearman's rank correlation analysis revealed a significant positive correlation ($R^2 = 0.79$, $P = 0.00003$) between copy number and the resistant phenotype (Fig 6B). This suggests that mosquitoes with more copies of the mutant allele have a higher chance of surviving exposure to PM.

## Discussion

While the World Health Organization recommends pirimiphos-methyl for IRS, the insecticide is not yet officially in use in Cameroon for vector control. In the present study, we have investigated the effectiveness of PM against mosquito populations in Cameroon, and assessed underlying resistance mechanisms particularly focusing on Ace-1 G280S target-site resistance. This study has revealed that while most populations remain susceptible to PM, there are nevertheless pockets of resistance, notably in *An. gambiae* s.s., that should be considered for the deployment of IRS intervention with organophosphates.

### 1- Contrasting resistance profile to organophosphates in malaria vectors in Cameroon

Populations from both *An. funestus* s.s. and *An. coluzzii* were mostly susceptible to organophosphates while those from *An. gambiae* s.s. were resistant. The full susceptibility to OP observed in *An. funestus* s.s. is in line with several reports across the continent consistently showing that populations of this species are fully susceptible to all organophosphates [39] and this is true even when escalation of resistance is reported against other classes notably pyrethroids [40–42]. The reason of such full susceptibility could lie with the predominant metabolic resistance mechanisms exhibited by this species through cytochrome P450s which are known to bioactivated organophosphates into their active form therefore making OP

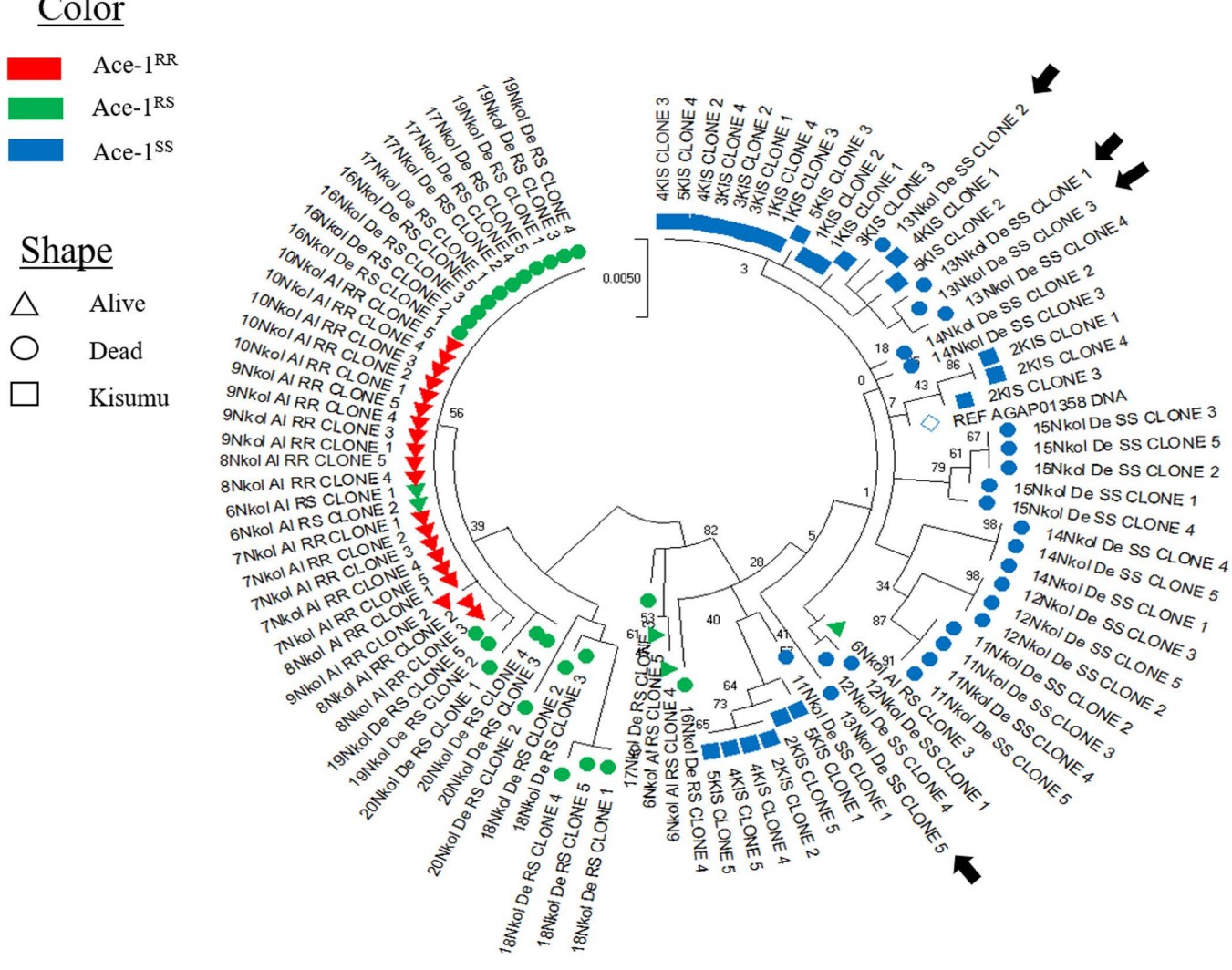

**Fig 5. Maximum-likelihood phylogenetic tree of Ace-1 gene from different clones.** The Tree presents two main clusters, Kisumu and dead SS on one side and alive and dead heterozygote on the other hand. Back arrows indicate 04 different haplotypes from clones of individual 13Nkolodom dead SS, sign of duplication.

more toxic to *An. funestus* s.s.. In contrast, *An. gambiae* s.s. relies on a more diverse array of resistance routes including a key role played by knockdown target-site resistance (kdr) [39] (Riveron et al 2018).

A general view of our results also revealed that mosquitoes exhibit greater susceptibility to PM compared to MA, aligning with similar trends observed on the toxicity effect of both insecticides in stored product pests [43] and on malaria vectors across Africa [10]. This difference occurs despite both being organophosphates. While the specific reasons remain unclear, potential contributing factors include pirimiphos-methyl's distinct mode of action, additional resistance mechanisms towards malathion, or variations in mosquito metabolic pathways. Notably, research suggests negative cross-resistance patterns based on the overexpression of specific cytochrome P450 enzymes [44,45], increasing or decreasing insecticidal activity depending on the relative rates of production of the active oxon form and inactive oxidative cleavage products respectively. Further investigations comparing these two organophosphates side by side are crucial to fully

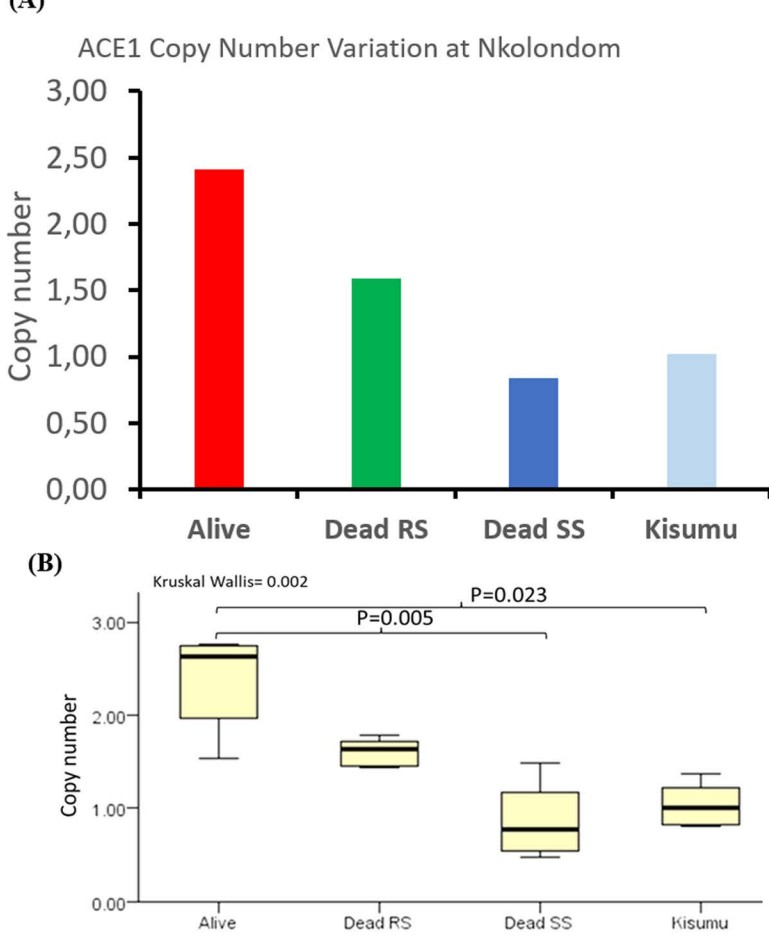

**(A)**

ACE1 Copy Number Variation at Nkolondom

**(B)**

**Fig 6. Relative copy number of Ace-1 gene among genotypes.** (A) Comparison of relative number of Ace-1 copies between individuals surviving pirimiphos methyl exposure (alive), and susceptible dead heterogenous, dead homogenous susceptible and the susceptible lab strain Kisumu. (B) Statistical test using Kruskal Wallis non-parametric tests.

understand the underlying mechanisms driving this differential susceptibility and guide future vector control strategies. In the context of Cameroon, the present study corroborates with previous findings that reported full susceptibility to organophosphate on *An. funestus* s.s. mosquitoes from Elende [33] and Mibellon [30], *An. coluzzii* from Gounougou [31], and resistance in Mangoum [29,46] and Nkolondom [32].

## 2- Ace-1ᴿ allele is a key driver of organophosphate resistance in *An. gambiae* s.s. from Cameroon

The highest level of resistance against organophosphates observed in *An. gambiae* s.s. populations compared to *An. funestus* s.s., could be attributed to the presence of Ace-1 G280S mutation. The correlation established between phenotypes and genotypes on mosquitoes exposed to PM and MA in Nkolondom further supports this observation. Moreover, the contribution of Ace-1ᴿ (280S) allele to resistance to organophosphates as well as carbamates is already well established in *An. gambiae s.l* [11,14,47,48], while this mutation is not present in *An. funestus* s.s. [17]. The Ace-1 N485I mutation linked with carbamate resistance [17] was absent in our study, strengthening the hypothesis of its limited geographic distribution to southern Africa [17].

The 3% frequency of the Ace-1$^R$ allele in *An. coluzzii* population from Njombe suggests a recent emergence of the mutation in this species in the locality. Three possible explanations are: 1) It could have spread from *An. gambiae* s.s. to *An. coluzzii* via introgression, as observed in West Africa [14], 2) It could have arisen independently, 3) It could have been introduced via migration. However, the presence of resistant allele among the few *An. gambiae* s.s captured during this study at Njombe, correlates the recent funding of Ngangue-Siewe et al. [28] in Cameroon, reporting in localities where *An. gambiae* s.s. and *An. coluzzii* cohabit, the presence of the mutation in both species.

### 3- Agricultural practices are key drivers of pirimiphos methyl resistance in *An. gambiae* s.s

The different levels of resistance observed in Nkolondom, Mangoum and Njombe might have been influenced by the local agricultural practices and the pesticide usage patterns [49,50]. The year-round cultivation of diverse crops like tomatoes, cabbages, parsley and lettuce, combined with novel wetland farming techniques like those employed in Nkolondom and Mangoum, has led to a surge in pesticide use. This practice, characterized by the creation of numerous water-retaining trenches from rainfall and irrigation, attracts *An. gambiae* s.s. mosquitoes for egg-laying (unlike *Anopheles funestus* s.l.). This exposes mosquito larvae to potential selection pressure from agricultural pesticides, contribute to the development of resistance [49,51]. Nwane and collaborator [49] reported the uncontrolled use by farmers of organophosphates in Mangoum and both organophosphates and carbamates in Nkolondom. Evidence of use of organophosphates by the farmers in both Mangoum and Nkolandom have been previously reported [52]. Moreover, the genetic diversity analysis of the Ace-1 G280S mutation, indicates a strong impact of pirimiphos-methyl exposure on the genetic diversity of the mosquito population from Nkolondom, likely due to positive selection for resistant genotypes. In fact, the low haplotype and nucleotide diversities in mosquitoes that survived exposure to PM compared to the dead ones suggests a strong selective sweep in the population. This is further supported by the negative and significant Tajima's and Fu's & Li's indexes, which indicate a reduction in genetic diversity consistent with recent positive selection. However, the sample size needs to be increased to draw more accurate conclusions. While Binyang and colleagues [46] reported higher values than our study, they also observed greater haplotype and nucleotide diversity among dead mosquitoes compared to alive ones exposed to carbamates in Cameroon.

### 4- Duplication of Ace-1 worsens resistance to PM in *An. gambiae* s.s

Despite matching Sanger sequencing results, TaqMan PCR used in this study on samples from Nkolondom exposed to PM, could not distinguish heterozygotes from homozygous resistant individuals due to potential Ace-1 gene duplication. Similar challenge leading to an inflation of RR and a reduced frequency of RS has been previously reported notably in Cameroon. As suggested by Bass et al. [34], adaptations may be required for TaqMan to identify duplicated genes. Yet, empirical evidence for such duplications remains limited. Various methods to assess Ace-1 duplication across mosquito species have been performed [53–55], but challenges lie in time consumption (requiring many crosses and bioassays) or cost (whole genome sequencing). Inspired by Essandoh et al. [11] and Ibrahim et al. [17], we explored an alternative approach in five mosquitoes with different TaqMan genotypes: cloning and sequencing a fragment of the Ace-1 gene. While less accurate than other methods, nevertheless it offers an indirect assessment of the presence of duplication. This approach revealed at least three different haplotypes in some individuals, confirming Ace-1 duplication in a field population of Nkolondom *An. gambiae s.s.*, particularly among susceptible individuals. This aligns with the high diversity observed in direct sequencing. However, the detection of less than three haplotypes as it is observed with alive homozygote resistant clones does not necessarily mean absence of duplication. In fact, besides duplication, qPCR reveals a higher number of copies in resistant mosquitoes compared to susceptible ones. Put together, results of duplication and CNV suggests that resistant mosquitoes to pirimiphos-methyl might harbor multiple, highly conserved copies of the Ace-1$^R$ allele. This aligns with previous reports of elevated copy number in resistant mosquitoes [14,56], the lack of nucleotide variation in resistant

homozygotes [11]. Dead RS individuals might be found with heterogenous duplication containing more susceptible copies than resistant, making them more vulnerable after exposure to insecticide. This is strengthened by the detection of many Ace-1$^S$ allele among clones of dead RS individuals. Supporting this, Grau Bové et al. [14] linked Ivorian *An. coluzzii* resistance to pirimiphos-methyl with Ace-1 duplications and multiple 280S alleles (resistant allele), showing increased survival with more 280S copies. Similarly, Assogba et al. [56] highlight the copy-number dependence, suggesting higher R copies for stronger resistance.

Based on our duplication evidence across all groups, mosquitoes can be renamed as: duplicated homozygous resistant R$^X$ (with x the varying number of copy), duplicated homozygous susceptible (S$^X$), and duplicated heterogenous (D) for Ace-1$^{RR}$, Ace-1$^{SS}$, and Ace-1$^{RS}$, respectively. While diverse nomenclatures are possible for heterogenous individuals based on variation of R/S allele numbers per chromosome [57,58]), whole-genome sequencing of these samples remains crucial for comprehensively characterizing copy number variation distribution, and their impact on Nkolondom's resistance.

## Conclusion

This study revealed a contrasting pattern of susceptibility among malaria vectors, with increasing resistance to PM in *An. gambiae* s.s. while *An. coluzzii* and *An. funestus* s.s exhibited near or full susceptibility. The distribution of Ace-1$^R$ allele correlated with its resistance profile, showing a high frequency in *An. gambiae* s.s., which is further exacerbated by the presence of copy number variation, whereas almost no resistant alleles were detected in the other species. The heterogeneity of OP resistance highlighted by this study underscores the necessity for monitoring the susceptibility of malaria vector populations before the large-scale implementation of IRS in Cameroon. A significant association found between Ace-1 and the resistant phenotype indicates that the distribution of this resistant allele should be evaluated nationwide, along with an investigation into the contribution of metabolic resistance, to help preserve the long-term efficacy of organophosphate-based interventions. Additionally, it is essential to evaluate the residual activity of insecticides prior to their selection for IRS.

## Supporting information

**S1 Fig: species identification according to sampling site.**
(PDF)

**S2 Fig: chromatogram presenting the position responsible for the Ace1-G280S mutation in *An. gambiae* s.s. from Nkolondom exposed to PM** . The figure highlights the presence of two peaks in one individual (the only one on the right) that was considered homozygote resistant with the TaqMan genotyping.
(PDF)

**S3 Fig: chromatogram presenting the position responsible for the Ace1-G280S mutation in *An. gambiae* s.s. from Nkolondom exposed to PM. (a)** 5 sequences of dead individuals that were genotyped as RR with the TaqMan, but present on the 3 readable sequences with two overlapping picks of A and G **(b)** 5 sequences of dead individual SS TaqMan genotype, presenting one pick of G on the 4 readable sequences. Useless sequences are those that exhibit multiple, overlapping peaks throughout the chromatogram. This overlap makes it difficult to confidently distinguish true heterozygotes from non-heterozygotes. They were then discarded.
(PDF)

**S4 Fig: scatter plot showing TaqMan genotyped results of mosquitoes from Nkolondom exposed to PM.** Two plots are visible, the one in red for "RR" mosquitoes and the one in blue for "SS" mosquitoes, no heterozygote was found. The selected samples for sequencing are identified with coding names. In the codes, A stands for Alive, and D for Dead phenotypes.
(PDF)

                                                    

**S5 Fig: chromatogram presenting the position responsible for the Ace1-G280S mutation.** In *An. gambiae* s.s. **(a)** and *An. coluzzii* **(b)** from Njombe, relative to his phenotype and TaqMan genotype. *An. gambiae* s.s. alive RR and *An. coluzzii* dead RS have the A pick for resistant allele.

(PDF)

We extend our sincere gratitude to all individuals who contributed to the collection of samples, particularly Drs Armel TEDJOU, Valdi DJOVA and Benjamin MENZE for providing mosquitoes respectively from Gounougou, Njombe, and Mibellon. We particularly acknowledge Mr Jalil NJIMBAM for his invaluable work on rearing and WHO bioassay testing. Additionally, we express our deep appreciation to Mrs Christelle KOPLONG for providing crucial assistance with laboratory work.

## Author contributions

**Conceptualization:** Judith Dandi-Labou, Jonas A. Kengne-Ouafo, Leon Mugenzi, Charles S. Wondji.

**Data curation:** Judith Dandi-Labou, Jonas A. Kengne-Ouafo, Magellan Tchouakui.

**Formal analysis:** Judith Dandi-Labou, Jonas A. Kengne-Ouafo, Leon Mugenzi.

**Funding acquisition:** Charles S. Wondji.

**Investigation:** Judith Dandi-Labou, Jonas A. Kengne-Ouafo.

**Methodology:** Judith Dandi-Labou, Leon Mugenzi, Charles S. Wondji.

**Project administration:** Murielle Wondji.

**Resources:** Murielle Wondji.

**Writing – original draft:** Judith Dandi-Labou, Charles S. Wondji.

**Writing – review & editing:** Judith Dandi-Labou, Jonas A. Kengne-Ouafo, Leon Mugenzi, Magellan Tchouakui, Murielle Wondji, Charles S. Wondji.

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
