## [Decision Letter · Decision Letter 0]

27 Dec 2024

PONE-D-24-54840Nationwide susceptibility profile of Anopheles and target site resistance mechanism against organophosphate in Cameroon.PLOS ONE

Dear Dr. Dandi-Labou,

Thank you for submitting your manuscript to PLOS ONE. After careful consideration, we feel that it has merit but does not fully meet PLOS ONE’s publication criteria as it currently stands. Therefore, we invite you to submit a revised version of the manuscript that addresses the points raised during the review process.

All reviewers felt that the draft manuscript has the potential to contribute to the existing literature in the subject area, though all have also raised important considerations that should be addressed in any subsequent versions of the manuscript. Please heed the suggestions raised by all reviewers regarding ways to improve the manuscript prior to submitting revised versions. 

We look forward to receiving your revised manuscript.

Kind regards,

James Colborn

Academic Editor

PLOS ONE

Journal Requirements:

“This research was funded by a grant from the Bill & Melinda Gates Foundation (Grant INV-006003) awarded to CSW. The views presented here are those of the authors and do not necessarily reflect the views of the BMGF.”

“This research was funded by a grant from the Bill & Melinda Gates Foundation (Grant INV-006003) awarded to CSW. The views presented in the manuscript are those of the authors and do not necessarily reflect the views of the BMGF.”

4. We note that Figures 1 & 2 in your submission contain [map/satellite] images which may be copyrighted. All PLOS content is published under the Creative Commons Attribution License (CC BY 4.0), which means that the manuscript, images, and Supporting Information files will be freely available online, and any third party is permitted to access, download, copy, distribute, and use these materials in any way, even commercially, with proper attribution. For these reasons, we cannot publish previously copyrighted maps or satellite images created using proprietary data, such as Google software (Google Maps, Street View, and Earth). For more information, see our copyright guidelines: http://journals.plos.org/plosone/s/licenses-and-copyright.

a. You may seek permission from the original copyright holder of Figures 1 & 2 to publish the content specifically under the CC BY 4.0 license. 

Reviewers' comments:

Reviewer's Responses to Questions

**Comments to the Author**

1. Is the manuscript technically sound, and do the data support the conclusions?

Reviewer #1: Yes

Reviewer #2: Partly

Reviewer #3: Yes

Reviewer #4: Yes

2. Has the statistical analysis been performed appropriately and rigorously? 

Reviewer #1: Yes

Reviewer #2: Yes

Reviewer #3: Yes

Reviewer #4: Yes

3. Have the authors made all data underlying the findings in their manuscript fully available?

Reviewer #1: Yes

Reviewer #2: Yes

Reviewer #3: Yes

Reviewer #4: Yes

4. Is the manuscript presented in an intelligible fashion and written in standard English?

Reviewer #1: Yes

Reviewer #2: Yes

Reviewer #3: Yes

Reviewer #4: Yes

5. Review Comments to the Author

Reviewer #1: The manuscript provided an insight into a countrywide resistance profile to organophosphates of two main malaria vectors in Cameroon and the related target site mechanism involved. Overall, the paper is relevant in its field and shows prospectives for decision making on organophosphates use in a context of pyrethroid resistance management. The methodology is in line with standards and the data generated was analysed in an appropriate way. The outcomes were discussed alongside with recent other studies from African malaria endemic settings.

Comments/Recommendations/Questions

Throughout the document: Replace “An. gambiae” by “An. gambiae s.s.” (when it is not referring to the complex). The use of “An. gambiae s.l.” is appropriate only while referring to the complex of species.

Line 30: Replace “WHO assays” by “WHO tube bioassays”

Line 40: Remove “moderately” as no resistance intensity bioassays were conducted against the mosquito population assessed.

Line 103-107: It looks like authors were highlighting the outcomes of the study. Better remove or rephrase as this is not the very best place for it.

Line 112: Replace “Anopheles” by “Anopheles”

Line 123-124: Replace "to insecticides” by “to the discriminating concentration of insecticides”.

Line 130: Replace “impregnated papers 0.25% PM,131 5% MA,” by “impregnated papers at discriminating concentration (0.25% PM, 5% MA)”

Line 132: The RH conditions used by authors overlap with the WHO-recommended range of 75% ± 10%, unlike temperature during which is supposed to be at 27 °C ± 2 °C (25-29°C). The 25 ± 1°C temperature slightly looked out of the WHO requirements.

Line 212-216: It is not clear whether species identification was performed on the whole mosquito size or a sub-sample. Kindly clarify in the manuscript.

Line 219: The number of mosquito specimens tested per location is not provided. It might be interesting to do add this on the graphs.

Line 224: At Njombe where vector population is composed of a mixture of An. colluzzii and An. gambiae ss, it would be preferable to present the dater per species.

Line 225: Replace “moderate resistance” by “resistance” (because no resistance intensity assay was conducted).

Line 231-236: Put in italics

Line 248: The way the pie charts are displayed around the map on Figure 2 requires improvement to avoid any sort of confusion. Better to have the stars replaced by the pie charts inside the map or not having a map at all with legend clear showing to what areas belongs the data.

Line 252-253: Put in italics

Line 256-258: Express the genotype in terms of proportion rather than number (61 individuals, 10 individuals, etc…)

Line 273-275: Express the genotype in terms of proportion as requested above.

Line 276: Replace “Ace” by “Ace-1”

Line 277-278: Put in italics

Line 283-285: Is there any justification of the mortality/genotype-based sampling used for Ace-1 gene polymorphism analysis? Same question for the sample size.

Line 334-338: Put in italics

Line 367-369: Put in italics

Line 382-384: Put in italics

Line 389: Replace “effectiveness” by “susceptibility”

Line 552-558: Reference number 10 is incomplete.

Reviewer #2: General comment

The authors addressed an important public health question in malaria research by investigating the effectiveness of organophosphate insecticides against mosquito populations circulating in six localities across Cameroon. They further assessed the implications of the genetic diversity of the Ace1 gene in the observed resistance patterns against these insecticides. Current recommended malaria intervention strategies, such as long-lasting insecticidal nets (LLINs) and indoor residual spraying (IRS), are insecticide-dependent, with IRS mostly relying on the use of organophosphates. However, although LLIN coverage is very optimal in Cameroon, the implementation of IRS is not yet effective. Studies conducted to monitor the susceptibility of local mosquito populations to insecticides prior to tool deployment are of significant public health interest for tool validation. The authors provided current data on the susceptibility profiles of three malaria vectors (An. gambiae s.s., An. coluzzii, and An. funestus) against malathion and pirimiphos-methyl, and the potential implications of Ace1 molecular target mutations in the observed resistance patterns. However, the volume of data produced does not support the conclusions of the study or the nationwide scope of the study. The data generated from sequencing and qPCR analysis of the Ace1 molecular target are not sufficient enough to draw confident statistical relationships about the phenotypic and genotypic resistance patterns of mosquito populations against malathion and pirimiphos-methyl in this study. Additionally, the authors should provide detailed information on the use of pirimiphos-methyl-related insecticides in Cameroon, as this is crucial for understanding the distribution of resistance alleles across the country or the targeted study sites. Furthermore, more details of the methodology used should be provided, along with measures taken to ascertain the accuracy and reproducibility of the data. Finally, strong evidences from the literature should be provided to support the study’s reported findings, and the applicability of IRS for malaria control in Cameroon, despite the observed increased resistance of OP insecticides. The full manuscript should be reviewed by native English speakers to correct any grammatical and vocabulary errors.

Specific comments are provided in each section of the main manuscript.

Title

The authors may consider rephrasing the manuscript’s title to reflect its content as “Susceptibility profiles and target site resistance mechanisms of Anopheles malaria vectors to organophosphates in Cameroon.” Although the study was carried out in six localities across five regions of Cameroon (5/10 regions), the volume of data produced in this manuscript does not support extrapolation of these findings to the whole country, and in some study sites as well. For instance, from the six targeted localities, only 20 mosquitoes were sequenced to assess the genetic diversity of the Ace1 gene across the sites, which is statistically not accurate.

Abstract

Line 32-33: replace “leading to…” with “necessitating the…”. The use of alternative insecticides due to escalating pyrethroid resistance is a necessity, rather than just a consequence of it.

Line 35: replace “such” with “these”

Line 36: replace “…resistance against…” with “…resistance to…”

Line 41: The sentence should be closed with a period/dot before starting the next sentence with "In contrast". If the use of the conjunction “both” refers to Anopheles gambiae s.s. and An. coluzzii, please specify this in the sentence, as in the previous sentence you talked about An. funestus and An. coluzzii instead.

Line 42: please define the meaning of PM and MA in the previous sentences (may be in line 36) where the full names of related insecticides are provided for the first time in the text.

Line 42: in the peri-urban. To link many ideas in the same sentence, commas should be added after the words Nkolondom and MA (46% mortality).

Line 45: replace “alignment’ with “association” and add a comma after the word profile.

Line 46: rephrase this sentence as … in the northern population of An. coluzzii, and …

Line 47: a higher frequency and not higher frequency

Line 50: add a comma after the word resistance

Lines 54-55: replace but with however and rephrase the sentence as follow: However, it will be necessary to consider the distribution of species and the development of resistance.

Introduction

Line 59-61: please use the new the recent world malaria report 2024 from the WHO to update the malaria burden data provided.

Line 63: IRS is not a tool. Replace “tools” by “intervention strategies”

Line 64-66: During the past decades, malaria control interventions have relied on four main classes of insecticides: organochlorines (since 1940s and 1950s for DDT), pyrethroids (since 1790s), organophosphates (sine 1960s), and carbamates (since 1970s) [ https://malariajournal.biomedcentral.com/articles/10.1186/s12936-018-2244-2 ].

Line 64: remove “their” before widespread.

Line 74: concerns have, not has

Line 79-80: With the cross-resistance mechanism existing between carbamates (CX) and organophosphates (OP), how can one associate the observed Ace-1 mutation specifically with OP and not CX? Differentiating between these is crucial for validating one of the insecticide classes for the development of control tools, such as IRS-based OP utilization.

Lines 96-99: …However, with the increasing use of OPs in agricultural settings, it is imperative to continuously monitor the presence and the extent of mosquitoes’ organophosphates resistance in the field, especially insecticides recommended by WHO for IRS. What is the extent of pirimiphos-methyl-based insecticide usage in agricultural settings in Cameroon, and how does this vary across the different malaria-endemic epidemiological zones? Can the authors provide detailed information on the use of pirimiphos-methyl-related insecticides in Cameroon, as this is crucial for understanding the distribution of resistance alleles across the country or the targeted study sites? Also, with the increased resistance to pirimiphos-methyl as reported by the authors (lines 91-92) in various countries implementing IRS, what other insecticides could be used for IRS, if this strategy is indeed very efficient in controlling mosquito vector expansion in endemic areas? These details should be provided for better understanding of the data reported in this manuscript.

Material and Methods

Mosquitoes sampling

Lines 114-118: is the lack of immature stages of Anopheles gambiae not due to the period of the study and/or site characteristics (market farming practices for instance)? When was the study carried out? The authors should provide details on the study period and the characteristics of the targeted sites, which are crucial for understanding the choice of each sampling technique used. Additionally, given the emphasis on the contribution of agricultural practices to organophosphate (OP) cross-resistance in public health (Anopheles), providing more details on the agricultural practices and main vector distribution in each locality could help better understand the data produced in this manuscript.

TaqMan genotyping

Lines 147-148: correct this sentence as …each reaction was conducted in 10ul total volume containing….

What QC measures were taken to validate the qPCR results?

Ace-a amplification and sequencing

Lines 165-166: From the 20 mosquitoes selected, how many were exposed dead and exposed survivors? Are the 20 mosquitoes from the same site or different sites? The authors should specify and provide more details on this. Finally, 20 mosquitoes seem not enough to validate the polymorphism of the Ace-1 gene in the mosquitos’ populations from up to 6 study sites. The authors should justify why only 20 mosquitoes sequenced, and raise this concern as a limitation to this study, as the observed genomic variations could not be extrapolated nationwide.

Line 168: rephrase as … a 50ul reaction mix containing 10pml of each primer …

Line 173-174: rephrase as … visualized on 10% agarose gel stained with Midori green.

Line 181: using default parameters of Tamura 3 model.

Line 182: …was built in network software. Specify the version of this online-based software.

Line 199: close the first brackets opened.

Lines 202-204: rephrase this sentence to complete the missing word. Each reaction mixture contained 1ul genomic DNA, 0.6uM of …..

Was the study not subjected to ethical and/or administrative considerations? The authors should clarify on this in a short subsection.

Results

Line 213: An. funestus constituted the main vector species in Elende and Mibellon, while An gambiae s.s constituted the main species in Mangoum and Nkolondom.

Lines 226-227: rephrase as: Alarmingly, mosquitos’ populations from Nkolondom exhibited greater resistance level to PM (80% mortality) and 46% (mortality).

Line 246: …in all An. funestus populations, supporting….

Line 256: From the XX individuals exposed to PM in Nkolondom, 61 exhibited the homozygous resistant allele RR, 10 exhibited the homozygous susceptible allele SS, and no heterozygous genotype (RS) was found in this population.

Line 258: if only 30/61 RR mosquitoes were alive, what was the status of the remaining 31 mosquitoes? if they were dead mosquitoes, could we infer that the distribution of the RR genotype is similar between alive and survivors’ mosquitoes from Nkolondom? Hence, to what extend could we associate the survivor status of these mosquitoes populations with the presence of the R allele?

Lines 260-261: It is written that “Mosquitoes with Ace-1R allele displayed a significantly higher ability to survive compared to those with the Ace-1S allele (OR=64.37, 95% CI=3.85–1075.38; P < 0.003)” – However, This seems not true if dead mosquitoes also carry the R allele at >50% (31/61) of the genotyped populations (see comment above, line 258).

Line 281: the author should consider removing the subtitle ‘Confirmation of genotyping result by direct sequencing’ from the text. The main title (Ace-1 gene polymorphism analysis) already explains the main idea of the following paragraph.

Lines 282-283: the authors should rephrase this sentence to correct grammatical and vocabulary errors, and fill in the missing words.

Line 289: two sequences and not two sequencing.

Line 298: Why choosing only 6 mosquitoes for sequencing from this site, giving that bioassays in this site were performed against MA and OP.

Line 344: Table presents…

Line 351: … the cloning helps…

Lines 371-379: How is the CNV data related to the genetic diversity of the Ace1 gene? Giving that the genetic diversity was higher in exposed dead mosquitoes, while the CNV is higher in exposed survivor mosquitoes.

Discussion

Line 430: Our and not your.

Lines 432-432: it is read that “The 3% frequency of the Ace-1 allele in An. coluzzii population from Njombe suggests a recent emergence of the mutation in this species in Cameroon” -

Line 456-457: The authors should explain the significance of the Tajima's and Fu's 457 & Li's indexes for better understanding of these data. Moreover, I am afraid that the number of samples sequenced (<20 sequences analyzed) in this study are not enough for a statistical validation of the data and to support the conclusions raised by the authors. The authors should be cautious is reporting such limited data. I will advice to mostly emphasize on the presence or not of the mutations in the study sites, and avoid any extrapolation that could bias the extend of the data to the entire study site or region or country.

459-460: What could be the consequences of the greater haplotype and nucleotide diversity observed among exposed dead mosquitoes after exposure to insecticides (OP or CM) on the efficacy of the IRS-based OP? given that the exposed survivors seem less diversified. The authors should provide further details on this to support the use of OP for IRS interventions in malaria endemic areas in short and long terms.

Line 462: the title reads “Duplication of Ace-1 worsens resistance to OP in An. gambiae in Cameroon” – However, sequencing of only five mosquitoes cannot generate enough data to support such conclusion. The authors should be modest in emphasizing about the strength of the data produced in their study. They should rephrase this title to report only the presence of duplicated copies of the Ace1 gene in the analyzed mosquitoes population (5) and recommend further investigations to assess it extend in a large mosquitoes populations from the same sites and other localities.

Conclusion

From the findings of this study, the authors should consider reporting the resistance profile of targeted mosquito species against malathion and pirimiphos-methyl, as well as the presence or absence of Ace1 mutations in the mosquito populations circulating in the study sites. The data generated from sequencing and qPCR analysis of the Ace1 molecular target are not sufficient enough to draw confident statistical relationships about the phenotypic and genotypic resistance patterns of mosquito populations against malathion and pirimiphos-methyl in this study.

Reviewer #3: The authors assessed the susceptibility profile of certain anopheline vectors, namely An. gambiae s.l. and An. funestus s.l., and their target site resistance mechanism to organophosphate insecticides in Cameroon. They suggest that malaria control by indoor residual spraying with organophosphates is a viable alternative in Cameroon, but that it will have to take into account the distribution of the species and the development of resistance. However, the authors need to make a number of corrections to improve the quality of their manuscript

Line 37 : ''...three malaria vectors (Anopheles gambiae s.s., An. coluzzii, and An. funestus s.l.)...''. I think the author should choose between the Anopheles complexes, in which case he replaces Anopheles gambiae s.s. and An. coluzzii with An. gambiae s.l. Otherwise, he should specify the molecular species within the An. funestus complex. Otherwise, it's confusing. Same of line lines 100 and 101. You should update the whole document, because in the manuscript you write An. gambiae without specifying the species or whether it is the complex. I know that others often prefer An. gambiae to An. gambiae s.s., but you weren't precise at the beginning.

Line 103-105: We revealed a... is this sentence necessary for this introductory section? it's not too clear and could be deleted or improved.

The author should ask a geographer to help him reproduce Figures 1 and 2. Otherwise these figures do not represent the work of a scientist.

Line 110: The author must specify the data collection period

Line 114 : ...Indoor aspiration method on adults' mosquitoes was used... In this case, the age of the mosquitoes will not be known and you will have to specify this as a limitation of the study in the discussion.

Line 130 : Capture by Indoor aspiration method will not give you the age of the mosquitoes

Line 219: An. gambiae s.l. or An. gambiae s.s.? You need to be consistent, otherwise in ‘Species identification’ above you referred to An. gambiae s.s.

As mentioned above, you should reproduce figure 2

Lines 373-374 : Alive mosquitoes had a significantly higher number of Ace-1 gene copies (2.4 ± 0.5) Line 374 compared to dead SS... You should specify whether these living mosquitoes were RR or RS or both

Lines 387-388: I suggest that this 1st sentence of the discussion be deleted as it adds nothing to the discussion at this point

Line 430: ...was absent in ‘your’ study... or ‘our’ study... ? Please make sure you correct the sentence so that it is understandable

Lines 432-435: 3% frequency and you're talking about emergence? I don't think so. Previous studies had shown 0% ? and by making a correlation you think that the difference is significant? If so, you should refer to studies carried out in the study area or in the country. It would be important to make corrections to this section

Reviewer #4: Titre : Nationwide susceptibility profile of Anopheles and target site resistance mechanism

against organophosphate in Cameroon.

General comments

Over the past decade, several studies have reported widespread resistance to pyrethroids and organochlorine insecticides, and sometimes local or suspected resistance to carbamates. Against this background, the search for new classes of insecticides effective against pyrethroid-resistant vectors has become a necessity. One possible solution is to use insecticides that are already available but have different vector targets to pyrethroids. Currently, non-pyrethroid insecticides pre-qualified by the WHO for indoor residual spraying belong broadly to three different classes (carbamates, organophosphates and neonicotinoids), with neonicotinoids joining the list in 2017 (WHO, 2020).

The spread of pyrethroid resistance in malaria vectors is threatening vector control, leading to the introduction of new insecticides such as neonicotinoids and pirimiphos-methyl (organophosphates) for indoor residual spraying (IRS). Monitoring mosquito susceptibility and understanding resistance mechanisms is essential for effective resistance management. It allows the selection of insecticides according to the control strategy to be implemented and the targeting of different types of LLINs or IRS insecticides according to the resistance mechanisms observed in an area.

This study assessed the level of resistance of populations of three species of Anopheles gambiae s.s., An. coluzzii and An. funestus s.l. to the insecticides pirimiphos-methyl and malathion in six locations in Cameroon. The study also identified potential mechanisms of resistance.

The results of this study showed that populations of An. funestus and An. coluzzii in Gounougou are completely sensitive to both organophosphates. On the other hand, An. coluzzii populations in Njombe showed a suspicion of possible resistance to malathion associated with a low frequency of the Ace-1R mutation (3%). Similarly, An. gambiae populations in Mangoum showed suspected resistance to pirimiphos-methyl (94% mortality) and confirmed resistance to malathion (50% mortality). However, An. gambiae populations in Nkolondom showed resistance to both pirimiphos-methyl (80% mortality) and malathion (46% mortality). This resistance is significantly associated with the Ace-1 mutation and resistance in this population. This resistance monitoring is crucial for the selection of an effective insecticide for IRS in Cameroon.

Overall, the manuscript is well written and easy to read. I agree with the publication of this manuscript provided that the following comments are taken into account:

Vectors are resistant to PM in several study sites.

1) Do you think it is appropriate to recommend this insecticide for IRS?

The duration of residual efficacy of insecticides applied to wall surfaces is one of the factors influencing the efficacy of indoor residual spraying. Several studies have shown that the remanence of PM is around 4 to 6 months (Chanda et al. 2013 in Zambia, Tchicaya et al., (2014) in Mbe in Côte d'Ivoire on banco and cement walls). On the other hand, a remanence of 8 months was observed by Haji et al. (2015).

2) Do you have any idea about the residual activity of PM in your context? If not, please specify the evaluation of residual activity of insecticides prior to the selection of IRS from the perspective of the study.

3) What is the duration of malaria transmission in the study area? Is it an area of perennial transmission? Mention the duration of transmission in the introduction.

If so, at least two rounds of IRS intervention per year are technically required. This requires considerable human and financial resources.

Overproduction of esterases has been associated with resistance to organophosphates and pyrethroids (Brogdon et al., 1999). It should be noted that increased expression of this enzyme may lead to detoxification of organophosphates.

4) Why haven't you done the enzyme test? If the results are available, include them in the results section of your study.

5) Do you think the Ace1 mutation is the only mechanism involved in the organophosphate resistance observed in Nkolondom?

New generation insecticides such as SumiShield® 50WG and Fludora® Fusion, recently approved by the WHO, are currently available and could be used to cover the entire duration of malaria transmission in regions with perennial malaria (Agossa et al., 2018a; Agossa et al., 2018b).

6) How sensitive are vectors to these new insecticides? It is now inconceivable to carry out a study of vector resistance without including the new insecticides.

6. PLOS authors have the option to publish the peer review history of their article (what does this mean? ). If published, this will include your full peer review and any attached files.

**Do you want your identity to be public for this peer review?** For information about this choice, including consent withdrawal, please see our Privacy Policy .

Reviewer #1: **Yes: ** Renaud Ines Segbegnon Govoetchan

Reviewer #2: **Yes: ** Francis Zeukeng

Reviewer #3: No

Reviewer #4: No

---

## [Author Response · Author response to Decision Letter 1]

10 Feb 2025

To the Editor in Chief of PLOS ONE

Dear editor,

Thank you for your time and consideration of our manuscript entitled “Nationwide susceptibility profile of Anopheles and target site resistance mechanism against organophosphate in Cameroon”, which was recently submitted to your journal. We are grateful to the reviewers for their thoughtful comments, which have significantly strengthened the manuscript.

We have carefully addressed their feedback and incorporated revisions within the text. As a major change, we have modified the title of the manuscript to suit our findings as suggested by the reviewer. New title: “Susceptibility profile of Anopheles and target site resistance mechanism against organophosphates in Cameroon.” Moreover, we have reproduced the two figures that necessitated to be refined. QGIS version 3.28.3 was used to generate the map using open access share files (https://gadm.org/). The manuscript has also been formatted according to PLOS ONS’s requirements: affiliations and levels heading have been formatted according to recommendations, and funding and competing interest information have been removed.

We are confident that the revised manuscript effectively addresses all feedback and now meets the rigorous quality standards for publication in PLOS ONE

Thank you again for your time and consideration. We look forward to hearing from you soon.

Sincerely,

Corresponding author

Judith Dandi-Labou

Reviewer comments to the Author

Reviewer #1:

Comments/Recommendations/Questions

Throughout the document: Replace “An. gambiae” by “An. gambiae s.s.” (when it is not referring to the complex). The use of “An. gambiae s.l.” is appropriate only while referring to the complex of species.

Thank you. An. gambiae has been replaced by An. gambiae s.s. when not referring to the complex throughout the document as suggested.

Line 39: Replace “WHO assays” by “WHO tube bioassays”

Ok, replaced. See line 39

Line 40: Remove “moderately” as no resistance intensity bioassays were conducted against the mosquito population assessed.

Ok, we have changed with “potentially resistant”. Lines 40- 41

Line 103-107: It looks like authors were highlighting the outcomes of the study. Better remove or rephrase as this is not the very best place for it.

OK, the sentence has been removed and sent to the conclusion section Lines 530-534

Line 112: Replace “Anopheles” by “Anopheles”

Ok. The word Anopheles has been italicized. Line 113

Line 123-124: Replace "to insecticides” by “to the discriminating concentration of insecticides”.

Ok. Replaced. Lines 133-134

Line 130: Replace “impregnated papers 0.25% PM,131 5% MA,” by “impregnated papers at discriminating concentration (0.25% PM, 5% MA)”

Ok. Replaced. Lines 140-141

Line 132: The RH conditions used by authors overlap with the WHO-recommended range of 75% ± 10%, unlike temperature during which is supposed to be at 27 °C ± 2 °C (25-29°C). The 25 ± 1°C temperature slightly looked out of the WHO requirements.

The reviewer is absolutely correct. During our bioassays, we recorded temperatures ranging from 24°C to 26°C, with an average of 25°C. While we did not observe any mortality in our control groups from the An. funestus s.s. population whose temperature was conducted at 24 °C, we acknowledge the importance of adhering strictly to WHO temperature recommendations. We appreciate your feedback and will ensure greater rigor in temperature conditions in future studies, in line with WHO guidelines.

The word “standard” have been removed. Line 142

Line 212-216: It is not clear whether species identification was performed on the whole mosquito size or a sub-sample. Kindly clarify in the manuscript.

Line 235-238

For populations known from previous studies to have only one primary vector species, we identified the species of just a subset of mosquitoes. However, in areas where multiple vector species are known to coexist (sympatric species), we performed species identification on all individual mosquitoes exposed to the insecticide.

We have clarified this in Material and Methods section. DNA extraction and species identification. Lines 151-156

Line 219: The number of mosquito specimens tested per location is not provided. It might be interesting to do add this on the graphs.

We tested as recommended by WHO (4 replicates of 25 mosquitoes each) 100 mosquitoes per location per insecticide, having 6 locations and two insecticides this represented 200 mosquitoes tested in each location, giving a total number 1200 adult females mentioned in line 243. Since this was not explicitly stated, we added the requested information in the manuscript line 243

Line 224: At Njombe where vector population is composed of a mixture of An. colluzzii and An. gambiae ss, it would be preferable to present the dater per species.

At Njombe, 10% of the 100 mosquitoes tested belonged to An gambiae s.s and 90% to An coluzzi, as mentioned in lines 238-240. An gambiae s.s. were few, and our targeted species were An. coluzzii. We removed them from the bioassay result

Line 225: Replace “moderate resistance” by “resistance” (because no resistance intensity assay was conducted).

Ok. “Moderate resistance” has been replaced by “possible resistance” following WHO recommendation. Line 250

Line 231-236: Put in italics

Lines 256-261 According to figure caption formatting, the legend is not in italics

https://journals.plos.org/plosone/s/figures

Line 248: The way the pie charts are displayed around the map on Figure 2 requires improvement to avoid any sort of confusion. Better to have the stars replaced by the pie charts inside the map or not having a map at all with legend clear showing to what areas belongs the data.

Ok, the figure has been reproduced. See Figure 2

Line 252-253: Put in italics

Lines 276-277 According to figure caption formatting, the legend is not in italics

https://journals.plos.org/plosone/s/figures

Line 256-258: Express the genotype in terms of proportion rather than number (61 individuals, 10 individuals, etc…)

OK. Proportions are added in brackets for more clarity. Lines 280-292

Line 273-275: Express the genotype in terms of proportion as requested above.

OK. Proportions are added in brackets for more clarity. Line 299-300

Line 276: Replace “Ace” by “Ace-1”

ok replaced Line 302

Line 277-278: Put in italics

Line 303-304 According to figure caption formatting, the legend is not in italics

https://journals.plos.org/plosone/s/figures

Line 283-285: Is there any justification of the mortality/genotype-based sampling used for Ace-1 gene polymorphism analysis? Same question for the sample size.

(Line 308-310) For cost purposes, we decided to confirm the genotyping on 20 individuals over 71 genotyped. Among the 20, we took 10 Alive and 10 dead Mosquitoes.

Being intrigued by the fact that many mosquitoes that were TaqMan RR died, we then divided the 10 dead among the two TaqMan genotypes we had (5 TaqMan RR and 5 TaqMan SS). The sequencing revealed that those “dead TaqMan RR” mosquitoes were heterozygotes. Having those sequences, we performed then the polymorphism analysis for a deeper understanding.

Line 334-338: Put in italics

Line 361-365 According to figure caption formatting, the legend is not in italics

https://journals.plos.org/plosone/s/figures

Line 367-369: Put in italics

Line 393-395 According to figure caption formatting, the legend is not in italics

https://journals.plos.org/plosone/s/figures

Line 382-384: Put in italics

Line 408-410 According to figure caption formatting, the legend is not in italics

https://journals.plos.org/plosone/s/figures

Line 389: Replace “effectiveness” by “susceptibility”

415-416 Since we are talking about the efficacity of the insecticide we think the word “effectiveness” is more appropriate than “susceptibility” which refers in this sentence to the response of the mosquitoes rather than to the action of the insecticide

Line 552-558: Reference number 10 is incomplete.

OK this have been adjusted. Line 596-597

USAID/President’s Malaria Initiative. Malaria Operational Plan FY 2018 and FY 2019; USAID/PMI:Cameroon, 2018.

Reviewer #2:

Title

The authors may consider rephrasing the manuscript’s title to reflect its content as “Susceptibility profiles and target site resistance mechanisms of Anopheles malaria vectors to organophosphates in Cameroon.” Although the study was carried out in six localities across five regions of Cameroon (5/10 regions), the volume of data produced in this manuscript does not support extrapolation of these findings to the whole country, and in some study sites as well. For instance, from the six targeted localities, only 20 mosquitoes were sequenced to assess the genetic diversity of the Ace1 gene across the sites, which is statistically not accurate.

The reviewer has raised a valid point about the limitations of extrapolating our findings to the entire country based on the current dataset. We agree that a more precise title would better reflect the content of our study.

The revised title now reads: " Susceptibility profile of Anopheles and target site resistance mechanism against organophosphates in Cameroon." Line 2

Abstract

Line 32-33: replace “leading to…” with “necessitating the…”. The use of alternative insecticides due to escalating pyrethroid resistance is a necessity, rather than just a consequence of it.

Ok changed see line 32

Line 35: replace “such” with “these”

Ok changed see line 35

Line 36: replace “…resistance against…” with “…resistance to…”

Ok changed see line 36

Line 41: The sentence should be closed with a period/dot before starting the next sentence with "In contrast". If the use of the conjunction “both” refers to Anopheles gambiae s.s. and An. coluzzii, please specify this in the sentence, as in the previous sentence you talked about An. funestus and An. coluzzii instead.

In contrast of the two Anopheles coluzzii populations, - where one was fully susceptible (the population from north) and the second displayed a possible resistance (the population from the south);- the two populations of An. gambiae ss of our study were resistant.

So we used “both” to mention the two An. gambiae s.s. populations (Mangoum and Nkolondom).

We have rephrased, changing “both” with “the two An. gambiae s.s. populations in this study were resistant…” see lines 41-45

Line 42: please define the meaning of PM and MA in the previous sentences (may be in line 36) where the full names of related insecticides are provided for the first time in the text.

Ok; done. See lines 36-37

Line 42: in the peri-urban. To link many ideas in the same sentence, commas should be added after the words Nkolondom and MA (46% mortality).

Ok, comma added. See lines 43 and 44

Line 45: replace “alignment’ with “association” and add a comma after the word profile.

Ok, replaced. See line 46

Line 46: rephrase this sentence as … in the northern population of An. coluzzii, and …

Ok, “the” has been added. See line 47

Line 47: a higher frequency and not higher frequency

Ok, “a” has been added. See line 48

Line 50: add a comma after the word resistance

Ok, added. See line 50

Lines 54-55: replace but with however and rephrase the sentence as follow: However, it will be necessary to consider the distribution of species and the development of resistance.

Ok, replaced and the sentence has been rephrased. See lines 53-54

Introduction

Line 59-61: please use the new the recent world malaria report 2024 from the WHO to update the malaria burden data provided.

Ok, the reference has been updated. See line 59-62

Line 63: IRS is not a tool. Replace “tools” by “intervention strategies”

Ok, replaced. See line 64

Line 64-66: During the past decades, malaria control interventions have relied on four main classes of insecticides: organochlorines (since 1940s and 1950s for DDT), pyrethroids (since 1790s), organophosphates (sine 1960s), and carbamates (since 1970s) [ https://malariajournal.biomedcentral.com/articles/10.1186/s12936-018-2244-2 ].

Line 64-67. Unfortunately except the DDT, we have not been able to find in the proposed article the other insecticide and date related so we didn’t totally rephrase as proposed by the reviewer Line 65

Line 64: remove “their” before widespread.

the sentence has been modified

- from “While pyrethroids currently dominate in LLINs due to their high mosquito killing efficacy and low mammalian toxicity, their widespread resistance among vectors poses a growing challenge to malaria control”

- To: ”While pyrethroids currently dominate in LLINs due to their high mosquito killing efficacy and low mammalian toxicity, the widespread resistance of mosquito vectors to this class of insecticide poses a growing challenge to malaria control”. Line 68-69

Line 74: concerns have, not has

Ok. Corrected. Line 77

Line 79-80: With the cross-resistance mechanism existing between carbamates (CX) and organophosphates (OP), how can one associate the observed Ace-1 mutation specifically with OP and not CX? Differentiating between these is crucial for validating one of the insecticide classes for the development of control tools, such as IRS-based OP utilization.

Line 80-81.

A way to associate the observed Ace-1 mutation to OP and not CX, could be to conduct bioassays with both carbamates and organophosphates on the same mosquito population and performing correlation tests between the presence of Ace-1R and the ability of mosquitoes to survive insecticide exposure. But in general the correlation is stronger in mosquitoes exposed to carbamates than to organophosphates. Ahoua Alou et al. (2010) discuss about reasons in their article titled ‘’Distribution of ace-1R and resistance to carbamates and organophosphates in Anopheles gambiae s.s. populations from Côte d'Ivoire.’’ (https://link.springer.com/article/10.1186/1475-2875-9-167), highlighting that it is a difficult question to address since many other parameters can be involved.

It should be noted that insecticide resistance is dynamic and complex, particularly regarding Ace-1, which exhibits copy number variations (CNVs). This resistance even varies among insecticides within the same class, as shown in this article, where the association of Ace-1R is stronger with Malathion than with Pirimiphos-methyl. One hypothesis arising from this work is that to survive Pirimiphos-methyl, mosquitoes might possess multiple well-conserved copies of the resistant allele. However, similar research needs to be conducted on mosquitoes exposed to Malathion for more accurate conclusions, since this article focused mainly on mosquitoes exposed to Pirimiphos-methyl to address CNV.

Lines 96-99: …However, with the increasing use of OPs in agricultural settings, it is imperative to continuously monitor the presence and the extent of mosquitoes’ organophosphates resistance in the field, especially insecticides recommended by WHO for IRS.

What is the extent of pirimiphos-methyl-based insecticide usage in agricultural settings in Cameroon, and how does this vary across the different malaria-endemic epidemiological zones?

Can the authors provide detailed information on the use of pirimiphos-methyl-related insecticides in Cameroon, as this is crucial for understanding the distribution of resistance alleles across the country or the targeted study sites?

We thank the reviewer for these questions which have substantially be addressed in the discussion, section entitled “3-Agricultural practices are key drivers of pirimiphos methyl resistance in An. gambiae “ Lines 470-477

Also, with the increased resistance to pirimiphos-methyl as reported by the authors (lines 91-92) in various countries implementing IRS, what other insecticides could be used for IRS, if this strategy is indeed very efficient in controlling mosquito vector expansion in endemic areas?

These details should be provided for better understanding of the data reported in this manuscript.

Some new classes of insecticides are positioned as valuable IRS insecticides against malaria vectors., Neonicotinoid (Acetamiprid, Clothianidin), Pyrroles (Chlorphenapyr) or Metadiamide (Broflanilide). But as proposed by WHO those insecticides, as well as PM, should be used in combi

---

## [Decision Letter · Decision Letter 1]

12 Mar 2025

Susceptibility profile of Anopheles and target site resistance mechanism against organophosphates in Cameroon.

PONE-D-24-54840R1

Dear Dr. Dandi-Labou,

We’re pleased to inform you that your manuscript has been judged scientifically suitable for publication and will be formally accepted for publication once it meets all outstanding technical requirements.

Kind regards,

James Colborn

Academic Editor

PLOS ONE

Additional Editor Comments (optional):

We have decided to accept the revised manuscript, though Reviewer 2 has requested a few small grammatical changes to the manuscript. Please ensure they are included with the final submission. See the specific request below.

"The authors have made the necessary corrections and I congratulate them on their effort.

However, in the methodology I didn't see a sentence that talked about the study period, i.e. the months or year when the study was carried out. Perhaps this is in an inappropriate place where I didn't pay attention. I would like the author to include it in the methodology.

With regard to Figure 1A, I noticed that the species are monocoloured on the study sites. Is it only the majority species that have been reported? Just a question of understanding"

Reviewers' comments:

Reviewer's Responses to Questions

**Comments to the Author**

1. If the authors have adequately addressed your comments raised in a previous round of review and you feel that this manuscript is now acceptable for publication, you may indicate that here to bypass the “Comments to the Author” section, enter your conflict of interest statement in the “Confidential to Editor” section, and submit your "Accept" recommendation.

Reviewer #2: (No Response)

Reviewer #3: (No Response)

2. Is the manuscript technically sound, and do the data support the conclusions?

Reviewer #2: Yes

Reviewer #3: Yes

3. Has the statistical analysis been performed appropriately and rigorously? 

Reviewer #2: Yes

Reviewer #3: Yes

4. Have the authors made all data underlying the findings in their manuscript fully available?

Reviewer #2: Yes

Reviewer #3: Yes

5. Is the manuscript presented in an intelligible fashion and written in standard English?

Reviewer #2: Yes

Reviewer #3: Yes

6. Review Comments to the Author

Reviewer #2: Although mosquito resistance to insecticides is dynamic, and despite the great susceptibility of Anopheles mosquitoes to organophosphates in Cameroon, the susceptibility profile to pyrimiphos-methyl is changing. Therefore, specifying the period of sample collection and bioassays is crucial for such a study. The authors should specify the exact period during which the study's activities were conducted.

Reviewer #3: The authors have made the necessary corrections and I congratulate them on their effort.

However, in the methodology I didn't see a sentence that talked about the study period, i.e. the months or year when the study was carried out. Perhaps this is in an inappropriate place where I didn't pay attention. I would like the author to include it in the methodology.

With regard to Figure 1A, I noticed that the species are monocoloured on the study sites. Is it only the majority species that have been reported? Just a question of understanding

7. PLOS authors have the option to publish the peer review history of their article (what does this mean? ). If published, this will include your full peer review and any attached files.

**Do you want your identity to be public for this peer review?** For information about this choice, including consent withdrawal, please see our Privacy Policy .

Reviewer #2: **Yes: ** Francis Zeukeng

Reviewer #3: No

---

## [Editor Report · Acceptance letter]

PONE-D-24-54840R1

PLOS ONE

Dear Dr. Dandi-Labou,

I'm pleased to inform you that your manuscript has been deemed suitable for publication in PLOS ONE. Congratulations! Your manuscript is now being handed over to our production team.

Kind regards,

on behalf of

Dr. James Colborn

Academic Editor

PLOS ONE